# APOBEC3B drives PKR-mediated translation shutdown and protects stress granules in response to viral infection

Lavanya Manjunath [1,2,3,11], Sunwoo Oh [1,2,3,11], Pedro Ortega [1,2,3], Alexis Bouin[2,4], Elodie Bournique [1,2,3], Ambrocio Sanchez [1,2,3], Pia Møller Martensen[5], Ashley A. Auerbach[6,7], Jordan T. Becker [7,8], Marcus Seldin[1,3], Reuben S. Harris[6,9], Bert L. Semler [2,4] & Rémi Buisson [1,2,3,10] ✉

Double-stranded RNA produced during viral replication and transcription activates both protein kinase R (PKR) and ribonuclease L (RNase L), which limits viral gene expression and replication through host shutoff of translation. In this study, we find that APOBEC3B forms a complex with PABPC1 to stimulate PKR and counterbalances the PKR-suppressing activity of ADAR1 in response to infection by many types of viruses. This leads to translational blockage and the formation of stress granules. Furthermore, we show that APOBEC3B localizes to stress granules through the interaction with PABPC1. APOBEC3B facilitates the formation of protein-RNA condensates with stress granule assembly factor (G3BP1) by protecting mRNA associated with stress granules from RNAse L-induced RNA cleavage during viral infection. These results not only reveal that APOBEC3B is a key regulator of different steps of the innate immune response throughout viral infection but also highlight an alternative mechanism by which APOBEC3B can impact virus replication without editing viral genomes.

APOBEC3 (Apolipoprotein B mRNA-editing enzyme catalytic polypeptide-like 3) is a family of polynucleotide cytosine deaminases that function as part of our innate immune system. APOBEC3 enzymes counteract various DNA and RNA viruses, retroviruses, and retrotransposons by inducing mutations in the virus genomes to stop their replication and protect cell integrity[1,2]. The APOBEC3 family of enzymes is encoded by seven genes (*A3A, A3B, A3C, A3D, A3F, A3G, and A3H*) located within the same region of chromosome 22 that have been amplified by tandem gene duplication during evolution[3]. All human APOBEC3 enzymes promote the deamination of cytosine to uracil in single-stranded DNA or RNA and have evolved different preferences for DNA or RNA sequences and structures to fight against diverse types of viruses that cells may encounter[4,5]. Many viruses such as human immunodeficiency virus type 1 (HIV-1)[6,7], hepatitis B virus (HBV)[8], rubella virus[9], Epstein–Barr virus (EBV)[10], and severe acute respiratory syndrome coronavirus 2 virus (SARS-CoV-2)[9,11] have been found to

[1]Department of Biological Chemistry, School of Medicine, University of California Irvine, Irvine, CA, USA. [2]Center for Virus Research, University of California Irvine, Irvine, CA, USA. [3]Center for Epigenetics and Metabolism, Chao Family Comprehensive Cancer Center, University of California Irvine, Irvine, CA, USA. [4]Department of Microbiology & Molecular Genetics, School of Medicine, University of California Irvine, Irvine, CA, USA. [5]Department of Molecular Biology and Genetics, Aarhus University, Aarhus C, Denmark. [6]Department of Biochemistry and Structural Biology, University of Texas Health San Antonio, San Antonio, TX, USA. [7]Institute for Molecular Virology, University of Minnesota – Twin Cities, Minneapolis, MN, USA. [8]Department of Biochemistry, Molecular Biology, and Biophysics, University of Minnesota – Twin Cities, Minneapolis, MN, USA. [9]Howard Hughes Medical Institute, University of Texas Health San Antonio, San Antonio, TX, USA. [10]Department of Pharmaceutical Sciences, School of Pharmacy & Pharmaceutical Sciences, University of California Irvine, Irvine, CA, USA. [11]These authors contributed equally: Lavanya Manjunath, Sunwoo Oh. ✉e-mail: rbuisson@uci.edu

accumulate APOBEC3-driven hypermutations in their genomes. Remarkably, APOBEC3 enzymes also protect cells against viruses independently of their DNA or RNA deaminase activity[12], emphasizing a conserved role for the APOBEC3 family that transcends the simple model of inducing mutations in viral genomes to stop their replication. A3G and A3F have been reported to interfere with HIV-1 reverse transcription[13–20], while A3H has been proposed to reduce HIV-1 replication possibly by binding to the HIV-1 genome near the primer-binding site[21,22]. Moreover, A3G and A3H suppress human T-cell leukemia virus type 1 (HTLV-1) through an editing-independent mechanism that has yet to be elucidated[23,24]. Similarly, A3B and A3F strongly inhibit LINE-1 retrotransposons independently of their DNA deaminase activity but the mechanism of suppression is also unknown[25–27]. Importantly, both deamination-dependent and independent mechanisms can work together to efficiently restrict viral replication. For example, A3G suppresses HBV through both its DNA deaminase-dependent and independent activity[28–30]. The different mechanisms by which APOBEC3 members suppress viral infection independent of genome editing are still poorly understood, highlighting a critical knowledge gap regarding how innate immune defense mechanisms protect host cells against viruses.

The first line of host defense mechanisms against viruses consists of the detection of pathogen-associated molecular patterns (PAMPs) by the host pattern-recognition receptors (PRRs)[31]. Key PRRs involved in PAMPs recognition are nucleic acid sensors that detect specific DNA or RNA sequences or structures present in viral genomes to trigger multiple signaling cascades converging on the production of type I interferons (IFNs), proinflammatory cytokines, and chemokines[31]. Double-stranded RNAs (dsRNAs) produced during viral replication and transcription are detected by the MDA5 or RIG-I dsRNA sensors that, in turn, stimulate MAVS leading to type I IFN production[32]. In addition, both protein kinase R (PKR) and ribonuclease L (RNase L) are activated by dsRNA which limits viral gene expression and replication through host shutdown of translation. PKR is a serine and threonine kinase that induces translation blockage and stress granule (SG) formation in infected cells through the phosphorylation of the translation initiation factor eIF2α[33,34]. In contrast, RNase L is activated by the dsRNA sensor OAS3 during viral infection[35] and suppresses translation by degrading both host and viral RNAs[36–38].

ADAR1 (Adenosine Deaminase Acting on RNA 1) is another deaminase enzyme in the human innate immune defense system that plays a critical function in preventing host-derived cytosolic dsRNAs from activating PRRs in the absence of viral infection[39,40]. ADAR1 binds dsRNA and catalyzes the deamination of adenosine to inosine (A-to-I)[41]. In uninfected cells, ADAR1 edits dsRNA, generating multiple mismatches that act to prevent MDA5 and PKR from binding to duplex RNAs[42]. ADAR1 is essential to limit self-dsRNA to initiate an MDA5-induced IFN response and to safeguard cells from translational shutdown mediated by PKR activation[40,42]. Consequently, mutations in ADAR1 cause Aicardi-Goutières syndrome (AGS), a rare genetic autoimmune disorder associated with chronic IFN production[43].

Similar to the APOBEC3 family, ADAR1 impacts viral replication through both deaminase-dependent and -independent mechanisms and can be either antiviral or pro-viral depending on the virus type and the cellular context[44]. Remarkably, many viruses such as Zika virus, measles virus, Sendai virus (SeV), or HTLV-1 take advantage of the ADAR1-mediated suppression of PKR, which protects the virus from any PKR-dependent antiviral responses, allowing its active replication in host cells[45–47]. Interestingly, the adenosine deaminase activity of ADAR1 is not required to block PKR stimulation during both Zika virus and HTLV-1 infections[46,47]. The dual requirements of cells to suppress aberrant PKR activation from self-dsRNA, while inducing PKR signaling pathway to block viral replication, highlight the presence of a fine-tuned regulatory mechanism. Yet, the mechanism by which infected cells counter-regulate ADAR1 pro-viral function(s) to activate PKR and restrict virus replication still needs to be elucidated.

In this study, we used different RNA viruses as tools to determine how A3B regulates innate immune signaling in response to viral infection. We identified A3B as a critical regulator of the RNA-activated innate immune sensor PKR. We found that A3B forms a complex with PABPC1 to promote PKR signaling pathways which mediate translational shutdown and SG formation of cells infected with diverse types of viruses. In addition, we demonstrate that A3B and PABPC1 counterbalance ADAR1's function in suppressing PKR activity, suggesting that A3B and PABPC1 are key factors of the innate immune response to limit ADAR1's pro-viral function. Moreover, we find that A3B is recruited to SGs through its interaction with PABPC1 to prevent SG destabilization and protect mRNAs associated with SGs from RNase L-induced RNA cleavage. We propose that A3B, in addition to its canonical role to edit viral genomes, functions with PABPC1 as important innate immunity mediators, protecting cells at different steps of the innate immune response against viral infections. These results highlight a complementary mechanism of how APOBEC3B protects cells against viruses without editing viral genomes.

## Results

### APOBEC3B is recruited to stress granules after SeV infection

To determine the mechanism by which A3B participates in the innate immune defense without editing viral genomes, we monitored A3B localization in cells infected with RNA viruses that cannot be directly edited by A3B-induced DNA deaminase activity. We initially focused on Sendai virus (SeV) as a tool to activate the innate immune response in host cells. SeV is a negative-sense, single-stranded RNA virus commonly used to understand the replication and pathogenesis of the *Paramyxoviridae* family that includes mumps, measles, Nipah, human parainfluenza virus 1 (HPIV-1), and respiratory syncytial viruses[48]. It is very challenging to monitor endogenous A3B localization by immunofluorescence due to low expression levels and because A3B antibodies cross-react with other APOBEC members, due to the strong sequence homology. To bypass this technical limitation, we developed DOX-inducible U2OS and A549 cell lines that express flag-tagged A3B at low levels without inducing any of the DNA damage typically observed in A3B overexpressing cells (Supplementary Fig. 1A)[49,50]. We selected U2OS and A549 cell lines that are both frequently used to study SeV infection and SGs[36,51–54]. Following SeV infection, we found that a fraction of A3B is re-localized to the cytoplasm of the cells to form multiple large structured foci (Fig. 1a, b). A well-characterized defense mechanism against viral infection is the formation of stress granules (SGs) that serve as antiviral signaling hubs and sequester host and/or viral mRNAs-proteins to modulate translation and reduce viral replication[55,56]. To determine whether A3B is localized in SGs after SeV infection, we monitored the formation of G3BP1 foci, a core constituent of SGs[55], and observed a strong colocalization with A3B in both U2OS and A549 cells (Fig. 1b and Supplementary Fig. 1a). We further confirmed A3B recruitment to SGs by monitoring the colocalization between A3B and TIA-1, FAM120A, and HuR, three other factors present in SGs (Supplementary Fig. 1C)[57,58]. In contrast, we did not detect any colocalization between A3B and EDC4, a marker for P-bodies (Fig. 1c)[57,59]. Together these results reveal that A3B localizes to SGs after SeV infection, suggesting that A3B may participate in the cellular antiviral defense mechanism(s) through SGs.

We next asked which domain is important for the recruitment of A3B to SGs. A3B is composed of two structurally similar domains[60]. The C-terminal domain (CTD) contains the DNA deaminase activity, and the N-terminal domain (NTD) is known to have a strong DNA and RNA-binding affinity and facilitates A3B enzymatic activity[61,62]. We removed the NTD (A3B^ΔNTD) or CTD (A3B^ΔCTD) domains and monitored A3B localization with G3BP1 following SeV infection. Deletion of the NTD, but not the CTD domain, completely abrogated the recruitment of A3B to SGs (Fig. 1d). In addition, we found that catalytically inactive A3B

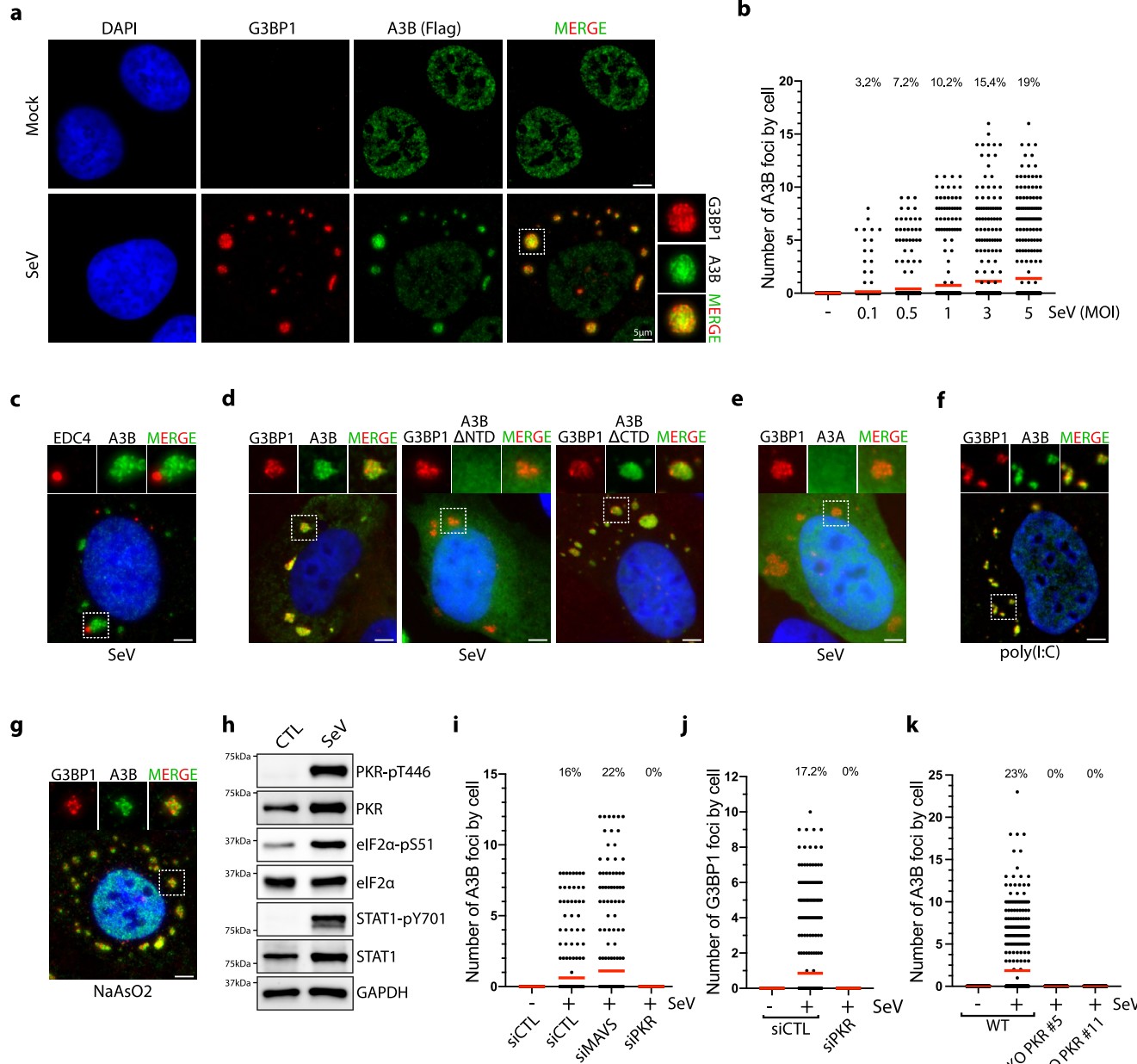

**Fig. 1 | SeV infection induces APOBEC3B localization to stress granules. a** U2OS-A3B-flag cells treated with DOX were infected with SeV (MOI = 1, 24hpi) and A3B and G3BP1 localization were monitored with a flag and a G3BP1 antibody respectively. Scale bar: 5 μm. **b** Quantification of the number of cytoplasmic A3B foci by cell in individual cells infected with SeV at the indicated MOI for 24 h. Top; percentage of cells with A3B foci. Red lines indicate the mean. (Number of cells, *n* = 500) **c** Immunofluorescence for A3B (Flag) and EDC4 in U2OS-A3B-flag cells treated with DOX and infected with SeV (MOI = 1, 24hpi). Scale bar: 5 μm. **d** U2OS cells were transfected with the indicated GFP tagged A3B constructs and subsequently infected with SeV (MOI = 1). A3B localization to SGs was monitored 24hpi by immunofluorescence. Scale bar: 5 μm. **e** The localization of A3A-GFP to SGs after SeV infection at MOI = 1 was monitored by immunofluorescence at 24hpi with GFP and G3BP1 antibodies respectively. Scale bar: 5 μm. Immunofluorescence for A3B

(Flag) and G3BP1 in U2OS-A3B-flag cells treated with DOX and transfected with poly(I:C) (200 ng/mL, 16 h) (**f**) or treated with NaAsO2 (250 μM, 1 h) (**g**). Scale bar: 5 μm. **h** U2OS cells were infected with SeV (MOI = 1), and indicated proteins and phosphorylation levels were monitored 24hpi by western blotting. **i** Quantification of the number of cytoplasmic A3B foci in individual U2OS cells knockdown for MAVS or PKR for 48 h and infected with SeV (MOI = 1, 24 hpi). Top; percentage of cells with A3B foci. Red lines indicate the mean. (Number of cells, *n* = 500). Quantification of the number of G3BP1 foci by cell in PKR knockdown cells (**j**) or cytoplasmic A3B foci by cell (**k**) in U2OS-A3B-Flag WT or PKR KO cells infected with SeV (MOI = 1, 24hpi). Top; Total percentage of cells with G3BP1 or A3B foci. Red lines indicate the mean. (Number of cells, *n* = 500). Source data are provided as a Source Data file.

mutant (A3B$^{E255Q}$) is also recruited to SGs (Supplementary Fig. 1D). As a control, we expressed A3A that is highly homologous to A3B$^{CTD}$ domain with 84.7% sequence identity[63]. Like A3B$^{ΔNTD}$, A3A was not recruited to the SGs (Fig. 1e). Taken together these results reveal that A3B recruitment to SGs is mediated through its NTD and that catalytic activity of A3B is not required for SG colocalization.

## PKR signaling-mediated stress granule formation is a prerequisite for A3B re-localization

Previous studies identified BORF2, expressed by Epstein–Barr virus, and ORF61, expressed by Kaposi's sarcoma herpesvirus, as interactors with A3B. These interactions lead to the re-localization and sequestration of A3B in the cytoplasm of the infected cells[10,64]. This suggests

that A3B subcellular localization is an important part of its antiviral activity and that viruses have evolved mechanisms of re-localization and sequestration of A3B. To establish whether A3B localization to SGs is mediated by the expression of viral proteins or by the host activation of the innate immune system-inducing SGs, we transfected cells with poly(I:C) (polyinosinic-polycytidylic acid). Poly(I:C) is a synthetic analog of dsRNA that is structurally similar to dsRNA present during viral infections and is therefore used to mimic the actions of RNA viruses[65]. Like SeV infection, poly(I:C) produces multiple A3B foci in the cytoplasm of cells colocalizing with G3BP1 (Fig. 1f). Furthermore, treatment with sodium arsenite (NaAsO$_2$) and H$_2$O$_2$, two drugs frequently used to induce the formation of SGs[66,67], also promotes A3B recruitment to G3BP1-associated SGs (Fig. 1g and Supplementary Fig. 1E). Therefore, A3B localization to SGs is not mediated by the expression of any viral protein but induced by signaling events resulting from viral infection and dsRNA sensing.

We next sought to identify the innate immune signaling pathway(s) responsible for A3B re-localization after SeV infection. Consistent with previous studies[65,68–70], SeV infection triggered STAT1 phosphorylation mediated by the RIG-I/MAVS-induced IFN signaling pathway along with both PKR and EIF2α phosphorylation, as evidence of activation of the PKR signaling pathway (Fig. 1h). To investigate which of these two PRRs are involved to induce SGs containing A3B, we knocked down MAVS or PKR in cells that were subsequently infected with SeV. Only in the absence of PKR were both A3B and G3BP1 foci formation abrogated (Fig. 1i, j and Supplementary Fig. 1F), which was further confirmed by knocking out PKR and subsequent SeV infection (Fig. 1k and Supplementary Fig. 1G). Taken together these results demonstrate that PKR-mediated SG formation during RNA viral infection is a required prerequisite for A3B re-localization to SGs.

## Identification of an APOBEC3B-interacting complex

To identify the interactors necessary for the recruitment of A3B to SGs, we expressed A3B-GFP/Flag in cells and immunoprecipitated A3B using an anti-Flag antibody followed by a GFP pulldown in the presence of DNase and RNase. RNase treatment functions to completely dissociate SGs or any RNA-dependent high molecular weight complex before the immunoprecipitation and thus minimize pulldown of proteins binding through RNA[54,61]. Simultaneously, we used SFB-GFP (S protein-FLAG-Strep binding peptide) as a control to identify nonspecific interactors. We separated the pulldown products by SDS-PAGE, excised protein bands from the gel, and analyzed them by mass spectrometry (MS) (Fig. 2a). Several highly-enriched proteins, including PABPC1, SAF-A, MATR3, DDX3, and MOV10, were identified in the A3B sample but not in the control sample (Fig. 2b and Supplementary Fig. 2A). Remarkably, three proteins interacting with A3B (PABPC1, DDX3, and MOV10) are well-known factors associated with SGs and have been found to modulate viral replication[57,71–74].

To validate the MS results, we performed an immunoprecipitation for A3B and confirmed the interaction with PABPC1, SAF-A, MATR3, DDX3, and MOV10 by western blot (Supplementary Fig. 2B). We did not detect any interaction between A3B and G3BP1, verifying that A3B immunoprecipitation did not precipitate the full SG structures. It is worth noting that PABPC1, DDX3, SAF-A, and MATR3 have been also identified in several published proteomic data sets of A3B pulldown from different cell lines[61,75,76]. Finally, we monitored the localization of these five A3B-interacting proteins after SeV infection via immunofluorescence. PABPC1, DDX3, and MOV10, but not SAF-A or MATR3 colocalized with A3B or G3BP1 in SGs (Fig. 2c). For this reason, we then focused our efforts on PABPC1, DDX3, and MOV10.

## PABPC1 recruits APOBEC3B to stress granules

After identifying the NTD as essential for A3B localization to SGs (Fig. 1d), we next tested whether the NTD domain is important for its interaction with PABPC1, DDX3, and MOV10. A3B lacking the NTD domain (A3B$^{ΔNTD}$) failed to pulldown PABPC1, DDX3, and MOV10 (Fig. 2d), suggesting that the interactions may be required to promote A3B localization to SGs. To further confirm the specificity of the interaction with A3B, we immunoprecipitated A3A and did not detect any interactions with PABPC1, DDX3, or MOV10 (Supplementary Fig. 2C). We next performed the reverse experiment by immunoprecipitating endogenous PABPC1. We found that DDX3 was pulled down with PABPC1 only in the presence of A3B, suggesting that A3B may act as a bridge connecting PABPC1 with DDX3 (Fig. 2e). Similarly, we performed an immunoprecipitation of endogenous DDX3 and again only detected PABPC1 when A3B was expressed (Supplementary Fig. 2D). Moreover, the knockdown of DDX3 did not affect the interaction between A3B and PABPC1, and inversely, PABPC1 knockdown did not affect the interaction between A3B and DDX3 (Supplementary Fig. 2E, F). Interestingly, MOV10 interacts with PABPC1 independently of the presence of A3B (Fig. 2e), implying that the interaction of MOV10 with A3B is indirect. Finally, the induction of SGs did not impact the interaction between A3B and PABPC1, DDX3, or MOV10 (Supplementary Fig. 3A, B). Thus, the multi-protein complex, consisting of A3B, PABPC1, DDX3, and MOV10 is pre-formed in cells and relocates to SGs after viral infection.

We then asked whether PABPC1, MOV10, and/or DDX3 are required for the recruitment of A3B to SGs. We individually knocked down PABPC1, MOV10, or DDX3 and monitored the localization of A3B to SGs after SeV infection, poly(I:C) transfection, or arsenite treatment. A3B localization with G3BP1 was strongly suppressed in the absence of PABPC1 in cells treated with arsenite or post poly(I:C), while MOV10 and DDX3 knockdown did not significantly impact A3B recruitment to SG-associated G3BP1 (Fig. 2f–h, and Supplementary Fig. 3C–F). Surprisingly, in the absence of PABPC1, the formation of G3BP1 foci was abrogated in SeV-infected cells but not after arsenite treatment or poly(I:C) transfection (Fig. 2i and Supplementary Fig. 3G, H), suggesting that PABPC1 has another important function in promoting SG formation exclusively during viral infection.

## APOBEC3B promotes PKR activity during viral infection

Arising from the observation that PABPC1 is essential for the formation of SeV-induced SGs (Fig. 2i), we then hypothesized that the PABPC1-A3B complex may play an important role in promoting the signaling events required for SG formation during viral infection. First, to determine whether A3B also impacts SG formation, we expressed A3B in SeV-infected cells and monitored the levels of SGs. A3B overexpression following DOX induction strongly enhanced both the number of G3BP1 foci formed in each cell and the percentage of cells positive for SGs (Fig. 3a). We then knocked down or knocked out endogenous A3B in U2OS or A549 cells. In the absence of A3B, both the number of SGs in cells and the percentage of cells positive for SGs decreased (Fig. 3b–d). Importantly, the levels of SGs were restored by complementing the knocked down cells with A3B (Fig. 3e), further suggesting that A3B is necessary for SG formation.

We next examined whether A3B promotes SGs in infected cells through the regulation of the PKR pathway, as PKR is essential for the formation of SeV-induced SGs (Fig. 1j). A3B overexpression during SeV infection strongly enhanced PKR and eIF2α phosphorylation (Fig. 3f and Supplementary Fig. 4A), whereas knockdown or knockout of endogenous A3B in U2OS or A549 cell lines compromised the cell's ability to activate PKR after SeV infection with about 4 to 6-fold decrease in PKR phosphorylation (Fig. 3g, h and Supplementary Fig. 4B, C). Consistently, the SKBR3 cell line that does not have the *A3B* gene failed to activate PKR and promote SGs after SeV infection (Supplementary Fig. 4D, E). While SKBR3 does not significantly express any APOBEC members, U2OS cells additionally expresses A3C and A3F at a relatively low level compared to A3B (Supplementary Fig. 4F). However, neither A3C nor A3F knockdown impacted the phosphorylation levels of PKR (Supplementary Fig. 4G–K). In addition, PKR

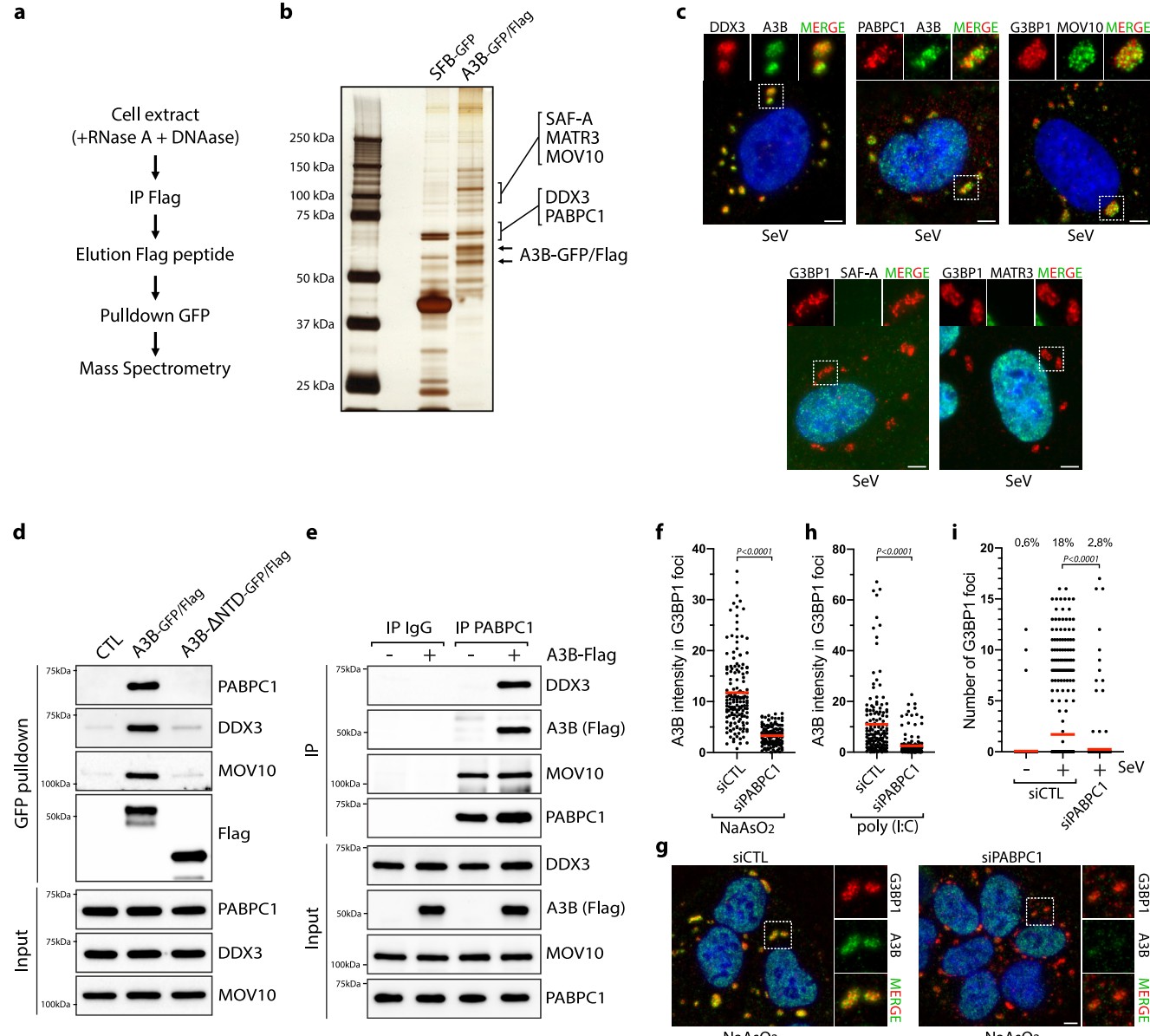

**Fig. 2 | PABPC1 recruits APOBEC3B to stress granules. a** A3B pulldown strategy. **b** Indicated pulldown samples from HEK-293T cells were analyzed by SDS-PAGE and silver staining. **c** DOX-treated U2OS-A3B-flag cells were infected with SeV (MOI = 1) and the cellular localization of the indicated proteins was monitored by immuno-fluorescence at 24hpi. Scale bar: 5 μm. **d** GFP pulldown of A3B-GFP/Flag or A3B-ΔNTD-GFP/Flag expressed in HEK-293T cells were analyzed by western blot with the indicated antibodies. **e** Immunoprecipitation of PABPC1 from U2OS-A3B-flag cells ± DOX and analyzed by western blot with the indicated antibodies. **f** Quantification of A3B intensity (arbitrary units) in G3BP1 foci in U2OS-A3B-flag cells treated with DOX and transfected with a siRNA control (CTL) or against PABPC1 for 40 h and then treated with NaAsO2 (500 μM, 1 h). Red lines indicate the mean (Number of G3BP1 foci, *n* = 150). *P*-values were calculated with a two-tailed Welch *t*-test.

**g** Immunofluorescence for A3B (Flag) and G3BP1 in U2OS-A3B-flag cells + DOX transfected with siCTL or siPABPC1 for 40 h and treated with NaAsO2 (500 μM, 1 h). Scale bar: 5 μm. **h** Quantification of A3B intensity (arbitrary units) in G3BP1 foci in U2OS-A3B-flag cells treated with DOX, transfected with a siRNA control (CTL) or against PABPC1 for 40 h, and then transfected with poly(I:C) (200 ng/mL, 16 h). Red lines indicate the mean (Number of G3BP1 foci, *n* = 150). *P*-values were calculated with a two-tailed Welch *t*-test. **i** Quantification of the number of cytoplasmic G3BP1 foci by cell in individual U2OS cells knocked down for PABPC1 and infected with SeV (MOI = 1, 24hpi). Top; percentage of cells with G3BP1 foci. Red lines indicate the mean (Number of cells, *n* = 500). *P*-values were calculated with a two-tailed Welch *t*-test. Source data are provided as a Source Data file.

activity was fully restored in A3B knockdown cells complemented with an siRNA-resistant A3B construct (Fig. 3i). The phosphorylation level of STAT1 (mediated by SeV-induced IFN response) was not affected by A3B expression (Fig. 3f), suggesting that A3B specifically regulates the PKR signaling pathway during viral infection. Importantly, PKR stimulation was independent of A3B DNA deaminase activity, indicated by the high level of PKR phosphorylation during either A3B or A3B$^{E255Q}$ expression and the absence of RNA editing events in SeV-encoded mRNAs (Supplementary Fig. 4L–N). To demonstrate that the DOX treatment used to induce A3B expression does not trigger SGs formation and PKR phosphorylation by itself, we DOX-treated a U2OS cell line which does not have the A3B expression inducible system incorporated into its genomes, and neither SG levels nor PKR phosphorylation were affected (Supplementary Fig. 5A, B).

Arsenite induces SGs independently of the PKR pathway (Supplementary Fig. 5C), in agreement with previous studies implicating HRI kinase activation after arsenite treatment[77]. Consistently, A3B expression did not increase the level of SGs after arsenite treatment

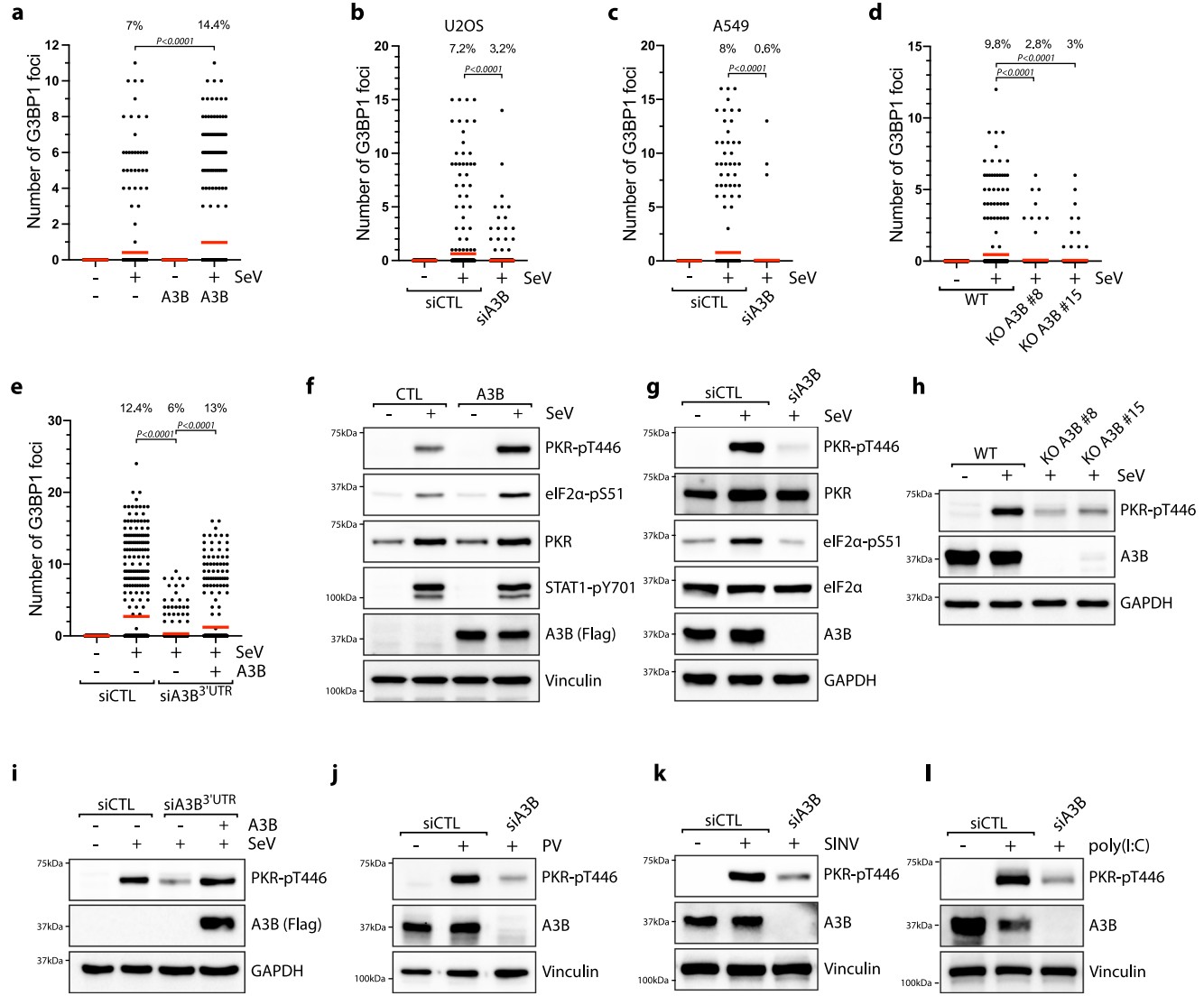

**Fig. 3 | APOBEC3B promotes PKR activity during viral infections.**
**a** Quantification of the number of G3BP1 foci by cell in U2OS-A3B-flag cells ± DOX and infected with SeV (MOI = 1, 24hpi). Top; percentage of cells with G3BP1 foci. Red lines indicate the mean (Number of cells, $n = 500$). *P*-values were calculated with a two-tailed Welch *t*-test. **b**, **c** U2OS or A549 cells were transfected with siRNA against A3B for 40 h and subsequently infected with SeV at MOI = 1. At 24hpi, the levels of G3BP1 foci were quantified by immunofluorescence. Top; percentage of cells with G3BP1 foci. Red lines indicate the mean (Number of cells, $n = 500$). *P*-values were calculated with a two-tailed Welch *t*-test. **d** Quantification of the number of G3BP1 foci in U2OS KO A3B cells infected with SeV (MOI = 1, 24hpi). Top; percentage of cells with G3BP1 foci. Red lines indicate the mean (Number of cells, n = 500). *P*-values were calculated with a two-tailed Welch *t*-test. **e** U2OS-A3B-flag cells ± DOX were transfected with a siRNA targeting the 3′UTR of endogenous A3B mRNA or a control (siCTL) for 40 h. The levels of G3BP1 foci were quantified by

immunofluorescence 24hpi with SeV at MOI = 1. Top; percentage of cells with G3BP1 foci. Red lines indicate the mean (Number of cells, $n = 500$). *P*-values were calculated with a two-tailed Welch *t*-test. **f** U2OS-A3B-flag cells ± DOX were infected with SeV (MOI = 1, 24hpi) and analyzed by western blot with antibodies against the indicated proteins. U2OS cells knocked down (**g**) or knocked out (**h**) for A3B were infected with SeV (MOI = 1, 24hpi) and analyzed by western blotting using the indicated antibodies. **i** U2OS-A3B-flag cells ± DOX were transfected with an siRNA targeting the 3′UTR of endogenous A3B mRNA or a control (siCTL) for 40 h. The levels of PKR-pT446, A3B, and GAPDH were detected by western blot 24hpi with SeV at MOI = 1. U2OS cells were transfected with siA3B for 40 h and then infected with either Poliovirus (PV) at MOI = 1 for 20 h (**j**), infected with Sindbis Virus (SINV) at MOI = 1 for 16 h (**k**), or transfected with 200 ng/mL poly(I:C) for 8 h (**l**). The levels of PKR-pT446, A3B, and Vinculin were analyzed by western blotting. Source data are provided as a Source Data file.

(Supplementary Fig. 5D), verifying that A3B modulates SG formation after viral infection specifically through the regulation of the PKR signaling pathway. To test whether PKR regulation by A3B is conserved for other types of RNA viruses, we selected poliovirus (PV) and Sindbis virus (SINV) –known to be detected by PKR[78,79] – along with the synthetic analog of dsRNA poly(I:C) as a surrogate for viral dsRNA[40]. Like SeV infection, A3B expression stimulates PKR activity after PV or SINV infection, or post poly(I:C) treatment, whereas knockdown of A3B abrogates PKR phosphorylation (Fig. 3j–l and Supplementary Fig. 5E–G). In addition, SKBR3 cells that do not express A3B failed to

activate PKR after both PV and SINV infections (Supplementary Fig. 5H) Thus, A3B has a critical function in regulating PKR during infection with different types of RNA viruses. Finally, we asked whether A3B impacts PKR-dsRNA condensates that are formed when dsRNAs are present in the cytoplasm[80,81]. We transfected cells with poly(I:C)-FITC to monitor PKR colocalization with dsRNAs and we confirmed that PKR formed large foci that are associated with poly(I:C) (Supplementary Fig. 5I). A3B knockdown decreased the number of PKR condensates, while A3B overexpression promotes their formation (Supplementary Fig. 5J, K), suggesting that A3B impacts PKR association with dsRNAs.

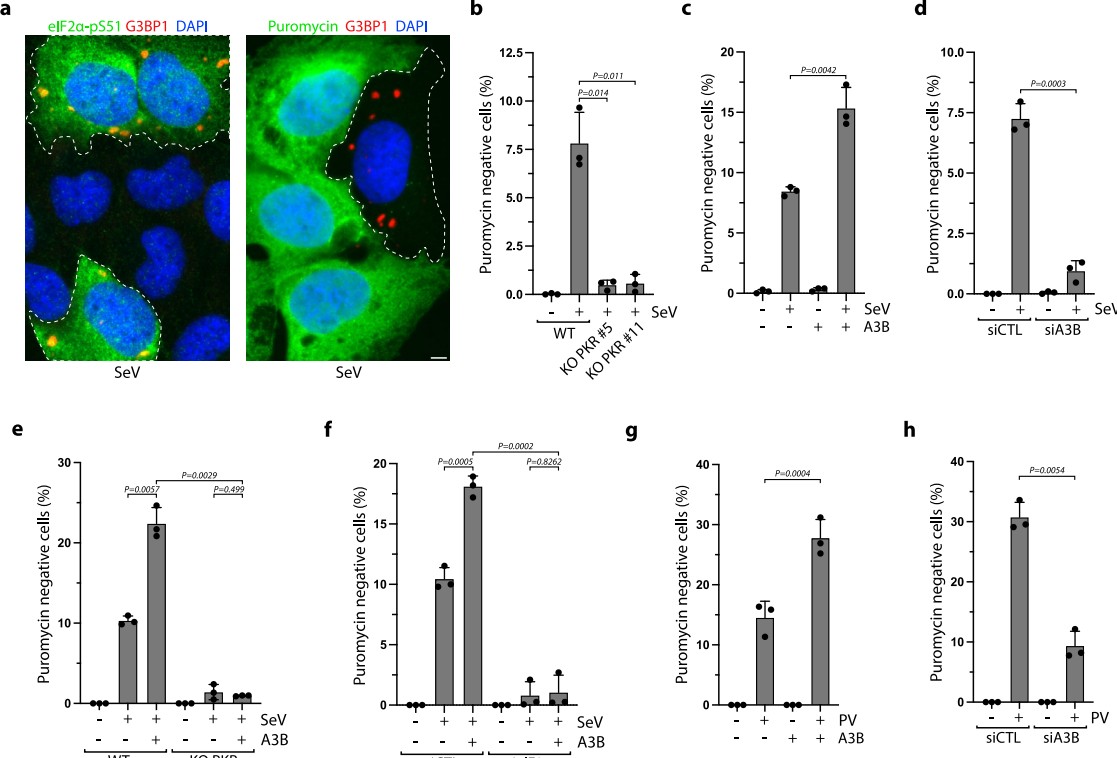

**Fig. 4 | A3B enhances PKR-mediated translation arrest. a** Representative immunofluorescence for G3BP1 and eIF2a-pS51 (left) or puromycin (right) in U2OS cells infected with SeV (MOI = 1, 24hpi). SG positive cells are marked with a dotted outline. Scale bar: 5 μm **b** Quantification of puromycin negative cells (%) in U2OS-A3B-Flag and U2OS-A3B-Flag PKR KO cells infected with SeV (MOI = 1, 24hpi). Mean values ± SD (Number of biological replicates, n = 3). P-values were calculated with a two-tailed Welch t-test. **c** Quantification of puromycin negative cells (%) in U2OS-A3B-flag cells ± DOX infected with SeV at 24hpi. Mean values ± SD (Number of biological replicates, n = 3). P-values were calculated with a two-tailed Welch t-test. **d** U2OS cells were transfected with siRNA against A3B for 40 h and subsequently infected with SeV at MOI = 1. At 24hpi, the levels of puromycin negative cells (%) were quantified by immunofluorescence. Mean values ± SD (Number of biological replicates, n = 3). P-values were calculated with a two-tailed Welch t-test. **e** U2OS-A3B-Flag WT or PKR KO cells ± DOX were infected with SeV at MOI = 1. At 24hpi, the levels of puromycin negative cells (%) were quantified by immunofluorescence. Mean values ± SD (Number of biological replicates, n = 3). P-values were calculated with a two-tailed Welch t-test. **f** Quantification of puromycin negative cells (%) in U2OS-A3B-flag cells ± DOX transfected with the indicated siRNA for 40 h and subsequently infected with SeV (MOI = 1, 24hpi). Mean values ± SD (Number of biological replicates, n = 3). P-values were calculated with a two-tailed Welch t-test. **g** U2OS-A3B-Flag cells ± DOX were infected with PV at MOI = 1. At 16hpi, the levels of puromycin negative cells (%) were quantified by immunofluorescence. Mean values ± SD (Number of biological replicates, n = 3). P-values were calculated with a two-tailed Welch t-test. **h** Quantification of puromycin negative cells (%) in U2OS cells knockdown for A3B and infected with PV (MOI = 1, 16hpi). Mean values ± SD (Number of biological replicates, n = 3). P-values were calculated with a two-tailed Welch t-test. Source data are provided as a Source Data file.

## APOBEC3B promotes PKR-mediated translation blockage

Activation of PKR induces the phosphorylation of eIF2α on serine 51, which blocks translation by preventing the transfer of Met-tRNAi to the 40S ribosomal subunit and thus promotes the dissociation of the translation initiation complexes from polysomes[82,83]. Translation shutdown is a critical defense mechanism for infected cells to limit the expression of viral proteins and virus replication and spread[34,82]. In addition to stopping the translation of new proteins, ribosome run-off following the arrest of translation releases RNA from polysomes, which allows RNA to act as a scaffold to nucleate RNA-binding proteins and initiate the formation of SGs[84]. Consistently, cells positive for SGs were also positive for eIF2α-pS51 (Fig. 4a), and eIF2α phosphorylation was completely abrogated in PKR KO cells infected with SeV (Supplementary Fig. 6A). To determine whether A3B promotes PKR-mediated translational shutdown, we measured levels of translation by labeling newly synthesized proteins with puromycin. Puromycin, a structural analog of aminoacyl tRNAs, is incorporated into nascent peptides by translating ribosomes[85]. Following SeV infection, cells positive for G3BP1 foci and PKR activity (i.e., eIF2α-pS51 positive cells) were negative for puromycin incorporation showing that these cells indeed maintained translational repression (Fig. 4a and Supplementary Fig. 6B).

In PKR KO cells, SeV-induced translation blockage was abrogated, as observed by the decrease in puromycin negative cells (Fig. 4b). Remarkably, A3B expression further increased the percentage of puromycin negative cells, whereas A3B knockdown reduced puromycin negative U2OS or A549 cells (Fig. 4c, d and Supplementary Fig. 6B, C). Importantly, A3B expression did not increase the percentage of puromycin negative cells in PKR KO cells or eIF2α knockdown cells (Fig. 4e, f), establishing that A3B-induced translation arrest is specifically mediated through the stimulation of the PKR-eIF2α pathway. Finally, we further confirmed these results in PV-infected cells. Similar to SeV, A3B expression stimulated translation blockage during PV infection, and A3B knockdown reduced the percentage of puromycin negative cells (Fig. 4g, h). Together, these results demonstrate that A3B is a critical factor to enhance translational blockage mediated by PKR during viral infection.

## APOBEC3B and PABPC1 counterbalance ADAR1-mediated PKR suppression

To determine whether the regulation of PKR is also mediated by PABPC1, we knocked down PABPC1 in SeV-infected cells. In the absence of PABPC1, PKR phosphorylation was strongly impaired

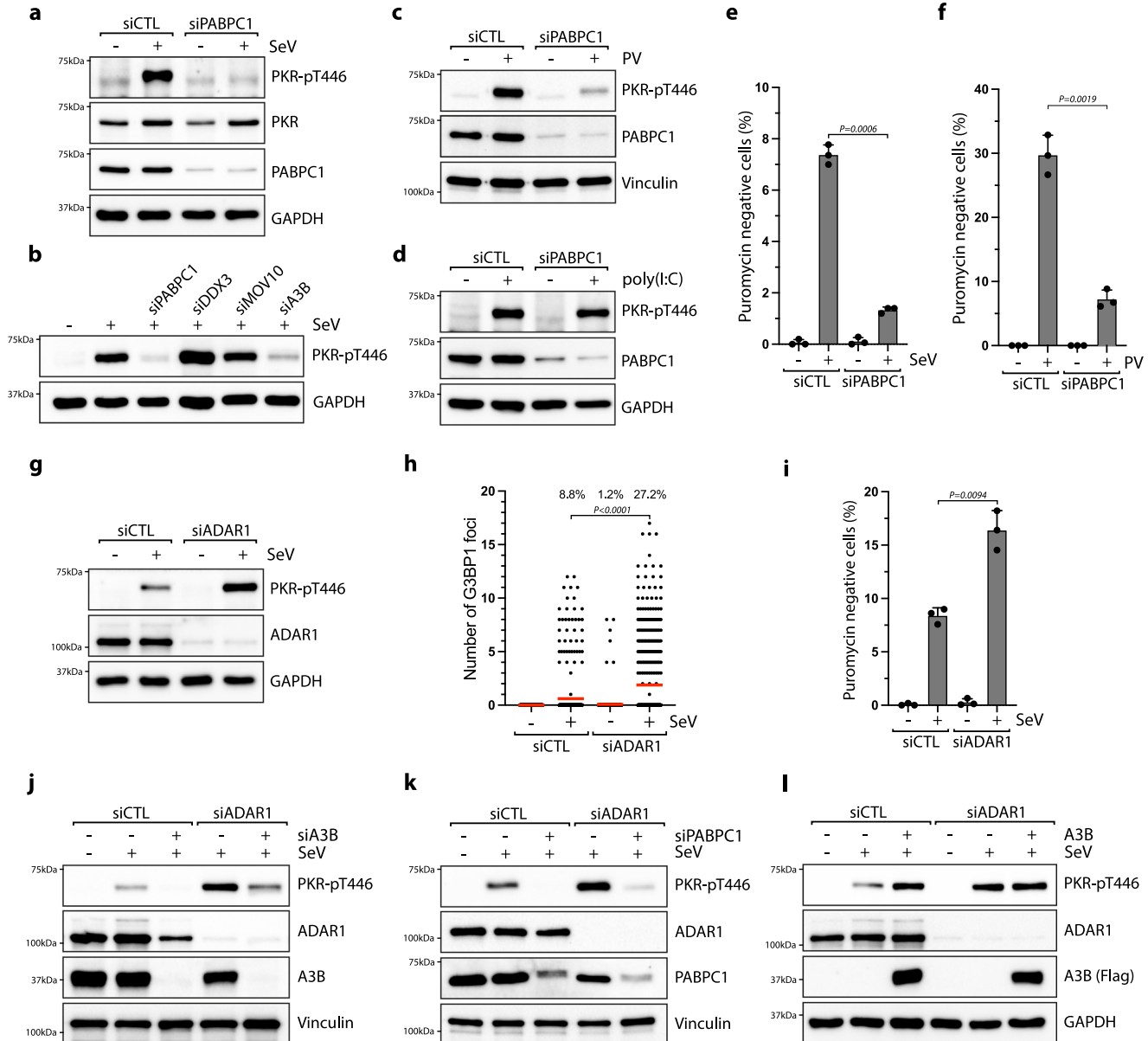

**Fig. 5 | APOBEC3B and PABPC1 counterbalance ADAR1 activity suppressing PKR. a, b** U2OS cells knocked down for 40 h with the indicated siRNAs were infected with SeV (MOI = 1, 24hpi) and the levels of PKR-pT446 and the indicated proteins were analyzed by western blotting. U2OS cells knocked down with a siRNA control (siCTL) or against PABPC1 for 40 h were infected with PV at MOI = 1 for 20 h (**c**) or transfected with 200 ng/mL poly(I:C) for 8 h (**d**). The levels of PKR-pT446, PABPC1, and vinculin were analyzed by western blotting. U2OS cells were transfected with siRNA against PABPC1 for 40 h and subsequently infected with SeV at MOI = 1 (**e**) or PV at MOI = 1 (**f**). At 24 hpi (SeV) or 16hpi (PV), the levels of puromycin negative cells (%) were quantified by immunofluorescence. Mean values ± SD (Number of biological replicates, *n* = 3). *P*-values were calculated with a two-tailed Welch *t*-test. **g–i**. U2OS cells knocked down with a siRNA control (siCTL) or against ADAR1 were infected with SeV at MOI = 1 for 24h. The levels of PKR-pT446 and the indicated proteins were analyzed by western blotting (**g**), and the number of G3BP1 foci in individual cells. (Red lines indicate the mean (Number of cells, *n* = 500)), (**h**) and the puromycin negative cells (Mean values ± SD (Number of biological replicates, *n* = 3)) (**i**) were quantified by immunofluorescence. *P*-values were calculated with a two-tailed Welch *t*-test. **j, k** U2OS cells knocked down with the indicated siRNAs were infected with SeV (MOI = 1, 24hpi) and the levels of PKR-pT446 were analyzed by western blotting. **l** U2OS-A3B-flag cells ± DOX and knocked down with siCTL or siADAR1 for 40 h were infected with SeV (MOI = 1, 24hpi) and the levels of PKR-pT446 and the indicated proteins were analyzed by western blotting. Source data are provided as a Source Data file.

(Fig. 5a). However, PKR activity was not suppressed by the knockdown of DDX3 or MOV10 (Fig. 5b), and we further confirmed the importance of PABPC1 promoting PKR activity in cells infected with PV (Fig. 5c). Interestingly, the absence of PABPC1 did not affect PKR phosphorylation after poly(I:C) treatment (Fig. 5d). As PABPC1 binds the 3′ poly(A) tract of mRNAs[86], these results suggest that 3′ poly(A) tract sequences present in viral RNAs but not in poly(I:C) are likely essential for PKR regulation by PABPC1. More importantly, PABPC1 knockdown strongly impaired translation arrest in both SeV and PV-infected cells (Fig. 5e, f),

further demonstrating that both PABPC1 and A3B are essential factors to enhance PKR-mediated translation shutdown in response to viral replication.

In contrast to A3B and PABPC1, ADAR1 acts as a pro-viral factor by preventing PKR activation from viral dsRNAs[41]. To investigate the interplay between ADAR1-mediated PKR suppression and A3B-PABPC1 complex promoting PKR activity, we knocked down ADAR1 and monitored PKR activity after SeV infection. In the absence of ADAR1, PKR phosphorylation strongly increased along with the level

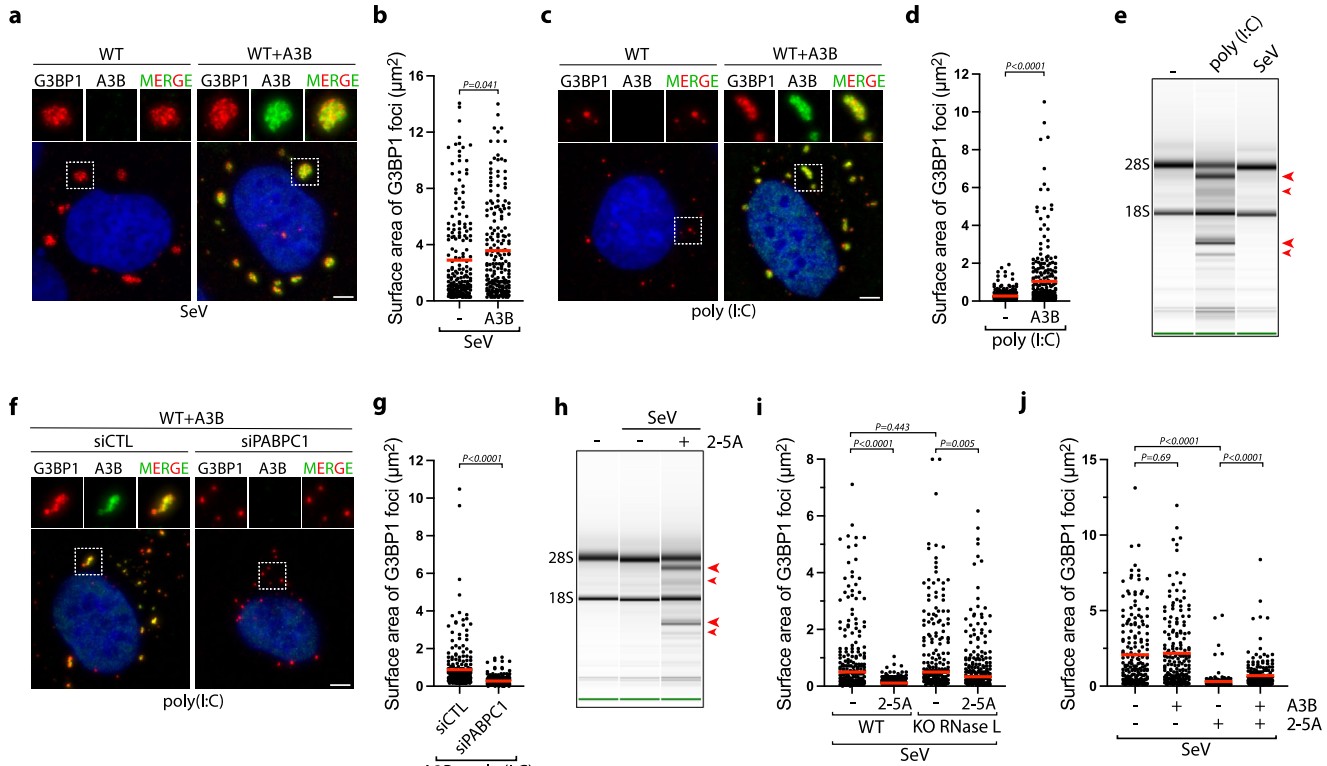

**Fig. 6 | APOBEC3B prevents RNase L-mediated stress granules disassembly.**
**a** U2OS-A3B-Flag cells ± DOX were infected with SeV (MOI = 1, 24hpi). A3B and G3BP1 localization were monitored by immunofluorescence with a flag or a G3BP1 antibody respectively. Scale bar: 5 μm. **b** Quantification of 200 G3BP1 foci surface area from the experiment described in (**a**). Red lines indicate the mean. *P*-values were calculated with a two-tailed Welch *t*-test. **c** Localization of G3BP1 and A3B were monitored by immunofluorescence in U2OS-A3B-Flag cells ±DOX after transfection with poly(I:C) (200 ng/mL, 16 h). Scale bar: 5 μm. **d** Quantification of 200 G3BP1 foci surface area from U2OS-A3B-Flag cells ±DOX transfected with poly(I:C) (200 ng/mL, 16 h). Red lines indicate the mean. *P*-values were calculated with a two-tailed Welch *t*-test. **e** Total RNAs were isolated and monitored for integrity by bioanalyzer after SeV infection (MOI = 1, 24hpi) or poly(I:C) transfection (400 ng/mL, 8 h). The red arrows indicated ribosomal RNA cleavage products. **f** U2OS-A3B-flag cells treated with DOX and knocked down with a siRNA control (siCTL) or against PABPC1 for 40 h were transfected with poly(I:C) (200 ng/mL, 16 h). G3BP1 and A3B

localization were monitored by immunofluorescence. Scale bar: 5 μm. **g** The surface area of 250 G3BP1 foci were quantified from the experiment described in (**f**). Red lines indicate the mean. *P*-values were calculated with a two-tailed Welch *t*-test. **h** U2OS cells treated with DOX were transfected with purified 2–5 A (10 μM) 8 h after infection with SeV (MOI = 1). 24hpi, total RNAs were collected and analyzed for integrity by bioanalyzer. The red arrows indicated ribosomal RNA cleavage products. **i.** Quantification of 200 G3BP1 foci surface areas were quantified in U2OS-A3B-flag or U2OS-A3B-flag-RNase L KO cells transfected with purified 2–5 A (10 μM) at 8hpi with SeV (MOI = 1). Cells were fixed and analyzed by immunofluorescence with G3BP1 and Flag antibodies at 24hpi. Red lines indicate the mean. *P*-values were calculated with a two-tailed Welch *t*-test. **j** U2OS-A3B-flag cells ± DOX transfected with purified 2–5 A (20 μM) at 8hpi SeV (MOI = 1). Surface area of G3BP1 foci (Number of G3BP1 foci, *n* = 200) was quantified at 24hpi. Red lines indicate the mean. *P*-values were calculated with a two-tailed Welch *t*-test.

of SGs and translation blockage (Fig. 5g–i). To further define the interplay between A3B-PABPC1 complex and ADAR1, we knocked down endogenous A3B or PABPC1 in SeV-infected cells, and both knockdowns strongly decreased PKR phosphorylation levels in cells lacking ADAR1 (Fig. 5j, k). Because A3B and PABPC1 regulate PKR phosphorylation regardless of ADAR1's activity, these results demonstrate that A3B- and PABPC1-induced PKR activation is not mediated through the direct inhibition of ADAR1 and functions through an independent pathway. We then overexpressed A3B in WT cells or cells knocked down for ADAR1. In both situations, PKR was phosphorylated at a similar level, but interestingly, A3B overexpression in cells depleted for ADAR1 did not further enhance PKR phosphorylation (Fig. 5l). These results suggest that A3B-induced PKR competes with ADAR1-mediated PKR suppression. An increase in A3B levels favors PKR activation but in the absence of ADAR1, a low level of A3B are now sufficient to fully promote PKR activity. We propose that the roles of A3B and PABPC1 during viral infection are to ensure the proper activation of PKR and translation shutdown in the presence of viral RNAs by counterbalancing ADAR1's inhibition of PKR activity.

## APOBEC3B protects stress granules from RNase L activity

We next analyzed whether A3B directly impacts the assembly or the structure of SGs. We monitored SG sizes in cells infected with SeV or transfected with poly(I:C). While SeV-infected cells promoted the formation of large G3BP1-associated SG structures that remained unchanged with A3B expression, poly(I:C) transfected cells induced small punctate G3BP1 foci that increased in size when A3B or A3B[E255Q] was expressed (Fig. 6a–d and Supplementary Fig. 7A, B). Poly(I:C) is known to activate RNase L in cells[35,87], an RNA endonuclease that cleaves host and viral RNAs, disturbing SG assembly, promoting SG disassembly, and inducing translation arrest to stop viral replication[36–38]. RNase L activity in cells can be readily detected by monitoring 18 S and 28 S ribosomal RNA degradation[35], and we verified that whereas poly(I:C) transfection stimulated RNase L activity, SeV infection did not (Fig. 6e, and Supplementary Fig. 7C, D). From these initial observations, we hypothesize that A3B plays a role in protecting SGs from RNase L activity.

To establish whether A3B promotes SG formation and/or prevents SG dissociation caused by the activity of RNase L, we first knocked down PABPC1 to suppress A3B's recruitment to G3BP1 foci in cells

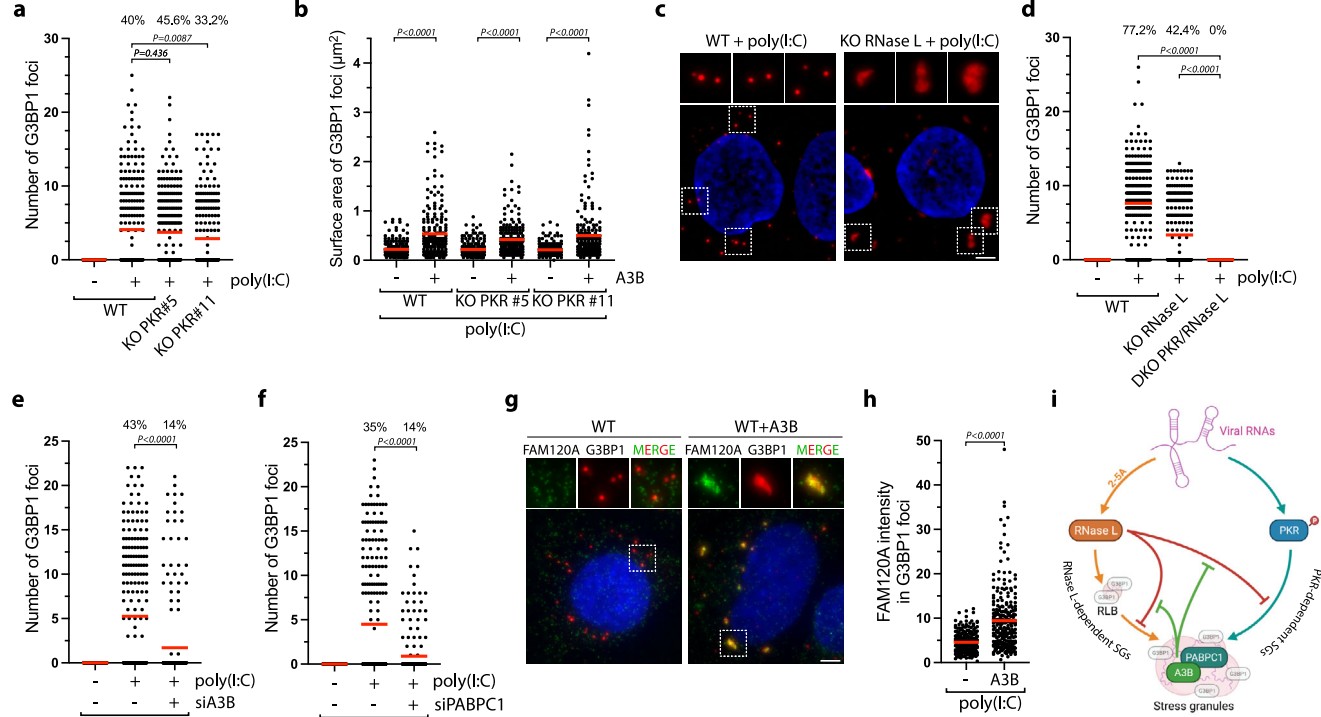

**Fig. 7 | APOBEC3B induces unique RNase L-dependent stress granules. a** U2OS WT and PKR KO cells were transfected with poly(I:C) (200 ng/mL, 16 h), and the number of G3BP1 foci by cell was quantified. Top; percentage of cells with G3BP1 foci. Red lines indicate the mean (Number of cells, $n = 250$). *P*-values were calculated with a two-tailed Welch *t*-test. **b** Quantification of G3BP1 foci area after transfection with poly(I:C) (200 ng/mL, 16 h) in U2OS-A3B-Flag WT and PKR-A3B-Flag KO cells. Red lines indicate the mean (Number of G3BP1 foci, n = 250). *P*-values were calculated with a two-tailed Welch *t*-test. **c**. Formation of G3BP1 foci after poly(I:C) (200 ng/mL, 16 h) transfection was monitored by immunofluorescence staining in U2OS-A3B-Flag WT and RNase L KO cells without DOX treatment. Scale bar: 5 μm. **d** U2OS-A3B-Flag WT, RNase L KO, PKR/RNase L DKO cells were transfected with poly(I:C) (200 ng/mL, 16 h), and the number of G3BP1 foci in individual cells were quantified without DOX treatment. Red lines indicate the mean (Number

of cells, n = 250). *P*-values were calculated with a two-tailed Welch *t*-test. Number of G3BP1 foci by cell was quantified in U2OS-A3B-Flag PKR KO cells without DOX treatment and knocked down with A3B (**e**) or PABPC1 (**f**) siRNAs for 40 h and transfected with poly(I:C) (200 ng/mL, 4 h). Red lines indicate the mean (Number of cells, n = 250). *P*-values were calculated with a two-tailed Welch *t*-test. **g** Localization of G3BP1 and FAM120A were monitored by immunofluorescence in U2OS-A3B-Flag cells ±DOX after transfection with poly(I:C) (200 ng/mL, 16 h). Scale bar: 5 μm. **h** Quantification of FAM120A intensity (arbitrary units) in G3BP1 foci from experiment described in (**g**). Red lines indicate the mean (Number of G3BP1 foci, $n = 250$). *P*-values were calculated with a two-tailed Welch *t*-test. **i** SG formation through an RNase L-dependent mechanism and a PKR-dependent mechanism during viral infection. A3B and PABPC1 protect SGs from RNase L-induced RNA cleavage.

transfected with poly(I:C) (Fig. 2h). In the absence of PABPC1, both A3B's colocalization with G3BP1 and the large sizes of G3BP1 foci were abrogated (Fig. 6f, g), suggesting that A3B recruitment to G3BP1 is essential to protect SGs against RNase L activity. We then artificially triggered RNase L activity in SeV-infected cells by transfecting the cells with purified 2′,5′-oligoadenylate (2–5A) (Fig. 6h)[88], a molecule produced by the dsRNA sensor OAS3 that activates RNase L by inducing its dimerization[35,89]. 2–5A transfection abolished the formation of large SGs and induced the formation of small punctate G3BP1 foci in an RNase L-dependent manner similar to puncta observed after poly(I:C) transfection (Fig. 6i and Supplementary Fig. 7E). Remarkably, A3B expression partially restored the size of G3BP1 foci post-2-5A transfection (Fig. 6j). To clearly demonstrate the role of A3B in preventing SG dissolution, we induced RNase L after SGs are formed by transfecting SeV-infected cells with poly(I:C) at 24hpi. Poly(I:C) transfection leads to a strong decrease in SG size but A3B expression limited RNase L-mediated SG dissolution (Supplementary Fig. 7F). We then performed the opposite experiment by transfecting cells with poly(I:C) first to induce RNase L before expressing A3B. A3B did not increase the size of SGs when expressed after RNAse L activation (Supplementary Fig. 7G). Together these results suggest that A3B protects SG structures against RNase L activity. However, once RNase L has degraded RNAs in the cells, A3B cannot promote G3BP1-RNA condensates and form large SGs due to the lack of scaffold RNAs present in the cells.

## APOBEC3B induces unique RNase L-dependent stress granules

In addition to promoting RNA decay and inhibiting canonical SG assembly, RNase L induces the formation of a unique ribonucleo-protein complex termed RNase L-dependent bodies (RLBs)[36,57]. RLBs are small, punctate G3BP1-positive foci that are generated independently of canonical dsRNA-induced SGs during viral infection because PKR activity is not required for RLB formation[36,57]. In accordance, the sizes of punctate G3BP1 foci formed post poly(I:C) transfection were not affected in PKR KO cells (Fig. 7a), and A3B expression increased G3BP1 foci size in both WT and PKR KO cells (Fig. 7b). In the absence of RNase L, G3BP1 formed foci that are larger and irregular in shape similar to canonical SGs. These G3BP1 foci were comparable to the ones formed after A3B expression (Fig. 7c), which were now abolished in RNase L and PKR double KO cells (Fig. 7d and Supplementary Fig. 8A). These results confirmed a previously reported model in which RNase L induces RLBs and suppresses PKR-dependent SGs[36,57]. Importantly, A3B knockdown in PKR KO cells decreases the level of G3BP1 foci post poly(I:C), while RNase L activity was not affected (Fig. 7e and Supplementary Fig. 8B). These data suggest that A3B promotes a unique type of RNase L-dependent SG. Likewise, knockdown of PABPC1 that is required to recruit A3B to G3BP1 foci also reduced the level of G3BP1 foci (Fig. 7f, and Supplementary Fig. 8B). Together, these results suggest that A3B is important for the stability of RNase L-induced G3BP1 foci in cells.

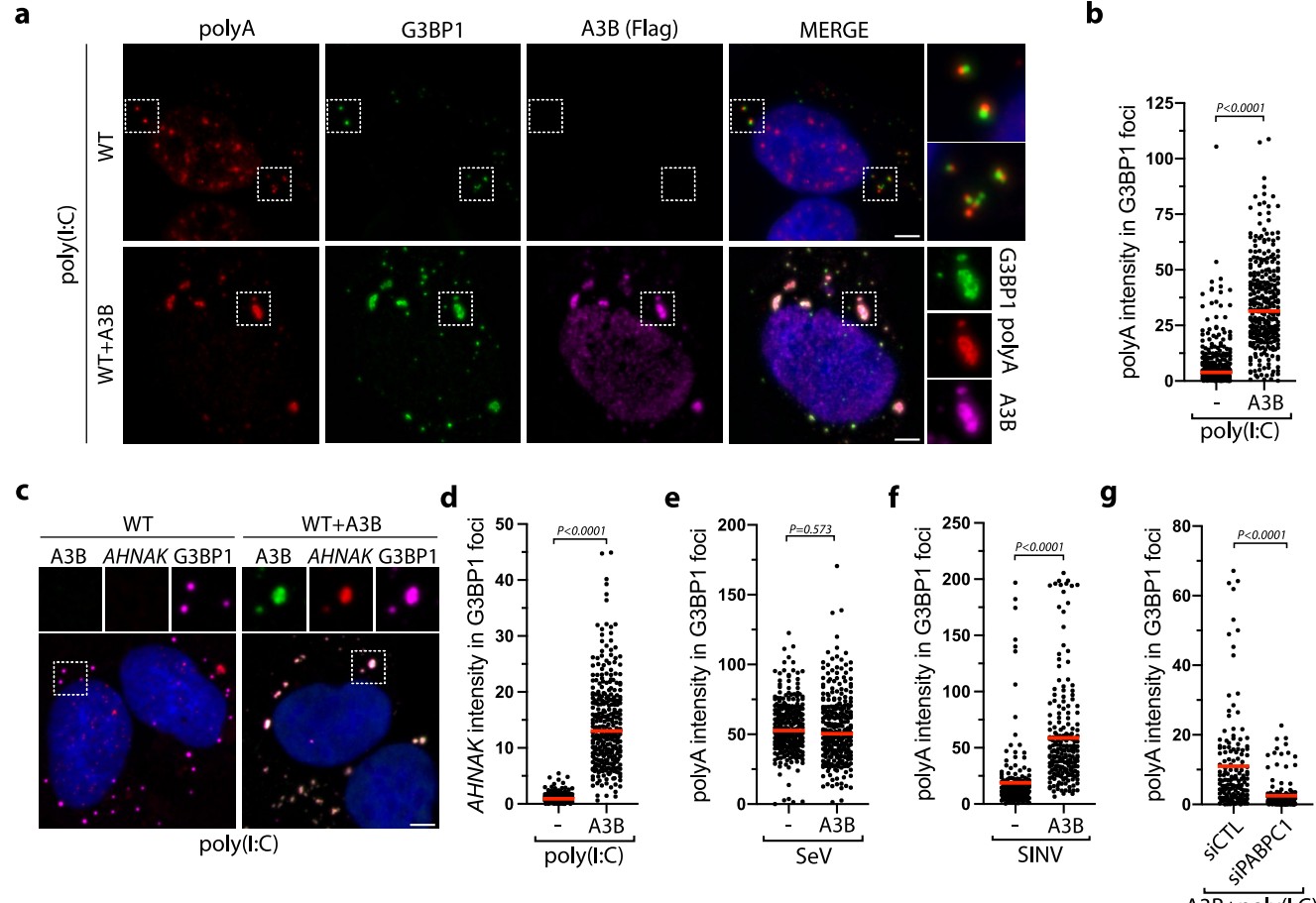

**Fig. 8 | APOBEC3B prevents the degradation of stress granule-associated RNAs by RNase L. a** Poly(A), G3BP1, and A3B localization were monitored by sequential IF-FISH (Fluorescence in situ hybridization) in U2OS-A3B-flag cells ± DOX after transfection with poly(I:C) (200 ng/mL, 16 h). Scale bar: 5 μm. **b** Quantification of poly(A) intensity in G3BP1 foci (arbitrary units) from the experiment described in (**a**). Red lines indicate the mean (Number of G3BP1 foci, $n = 300$). *P*-values were calculated with a two-tailed Welch *t*-test. **c** A3B, *AHNAK* mRNA, G3BP1 localization were monitored by sequential IF-FISH in U2OS-A3B-flag cells ± DOX after transfection with poly(I:C) (200 ng/mL, 16 h). Scale bar: 5 μm. **d** Quantification of *AHNAK* mRNA intensity (arbitrary units) in G3BP1 foci after the treatment described in (**c**). Red lines indicate the mean (Number of G3BP1 foci, $n = 300$). *P*-values were calculated with a two-tailed Welch *t*-test. Poly(A) intensity (arbitrary units) in 200 G3BP1 foci was quantified in U2OS-A3B-flag cells ±DOX subsequent to SeV infection (MOI = 1, 24 hpi) (Number of G3BP1 foci, $n = 200$) (**e**) or SINV infection (MOI = 1, 24 h) (Number of G3BP1 foci, $n = 300$) (**f**). Red lines indicate the mean. *P*-values were calculated with a two-tailed Welch *t*-test. **g** U2OS-A3B-flag cells + DOX were knocked down with indicated siRNAs for 40 h and transfected with poly(I:C) (200 ng/mL, 16 h). The intensity of poly(A) (arbitrary units) in G3BP1 foci was quantified. Red lines indicate the mean (Number of G3BP1 foci, $n = 150$). *P*-values were calculated with a two-tailed Welch *t*-test.

We then hypothesized that A3B localization to RNase L-dependent G3BP1 foci affects their protein composition. We first monitored FAM120A, a specific maker of canonical SGs that is not present in RLBs[57]. We found that large G3BP1 foci induced by A3B expression contain FAM120A (Fig. 7g, h). Similarly, A3B promotes HuR recruitment to G3PB1 post poly(I:C) (Supplementary Fig. 8C). Thus, A3B not only promotes an increase in G3BP1 foci size but also influences the protein composition of G3BP1-associated foci. Next, we expressed A3B in cells depleted for G3BP1 and G3BP2 (Supplementary Fig. 8D), which are essential for the formation of SGs but not RLBs[57,90]. In the absence of G3BP1 and G3BP2, the size of A3B foci was significantly reduced along with FAM120A and HuR recruitment in WT cells or cells knocked down for PKR post poly(I:C) (Supplementary Fig. 8E–H). Taken together, these results revealed that A3B converts RNase L-dependent bodies (RLBs) to unique RNase L-dependent SGs (RLSGs) independently of PKR (Fig. 7i).

### APOBEC3B protects stress granule-associated RNAs from RNase L activity

RNAs serve as molecular scaffolds recruiting multivalent RNPs and are important components for the formation of higher-ordered SG structures[91–93]. To investigate the mechanism by which A3B converts RLBs to RLSGs by protecting SG structures form RNase L cleavage activity, we monitored mRNA recruitment to G3BP1 foci by FISH. We first used a poly(dT) probe which binds to 3' poly(A) tracts to detect mRNA recruitment. Many G3BP1 foci induced post poly(I:C) were found to be closely associated with punctate poly(A) signals, but not fused with each other. Rather, G3BP1 and poly(A) foci remained distinct condensates in cells (Fig. 8a). In contrast, we detected a strong poly(A) signal colocalizing with G3BP1 in A3B-induced RLSGs (Fig. 8a, b). We further confirmed this result by monitoring the recruitment of *AHNAK* mRNA, an mRNA that is specifically enriched in SG structures[57,72]. *AHNAK* mRNAs were not present in RLBs post poly(I:C) but were highly enriched in A3B-induced RLSGs (Fig. 8c, D). However, we did not detect any changes in mRNA level recruitment in SGs in SeV-infected cells that do not trigger RNase L (Fig. 8e), suggesting that the A3B's mRNA protection function is necessary only when RNase L is activated. Furthermore, we infected cells with SINV which induced RNase L activation (Supplementary Fig. 9A)[35] in contrast to SeV (Fig. 6e). In SINV-infected cells, A3B expression likewise promoted an augmentation of poly(A) signals in G3BP1-associated SG (Fig. 8f and

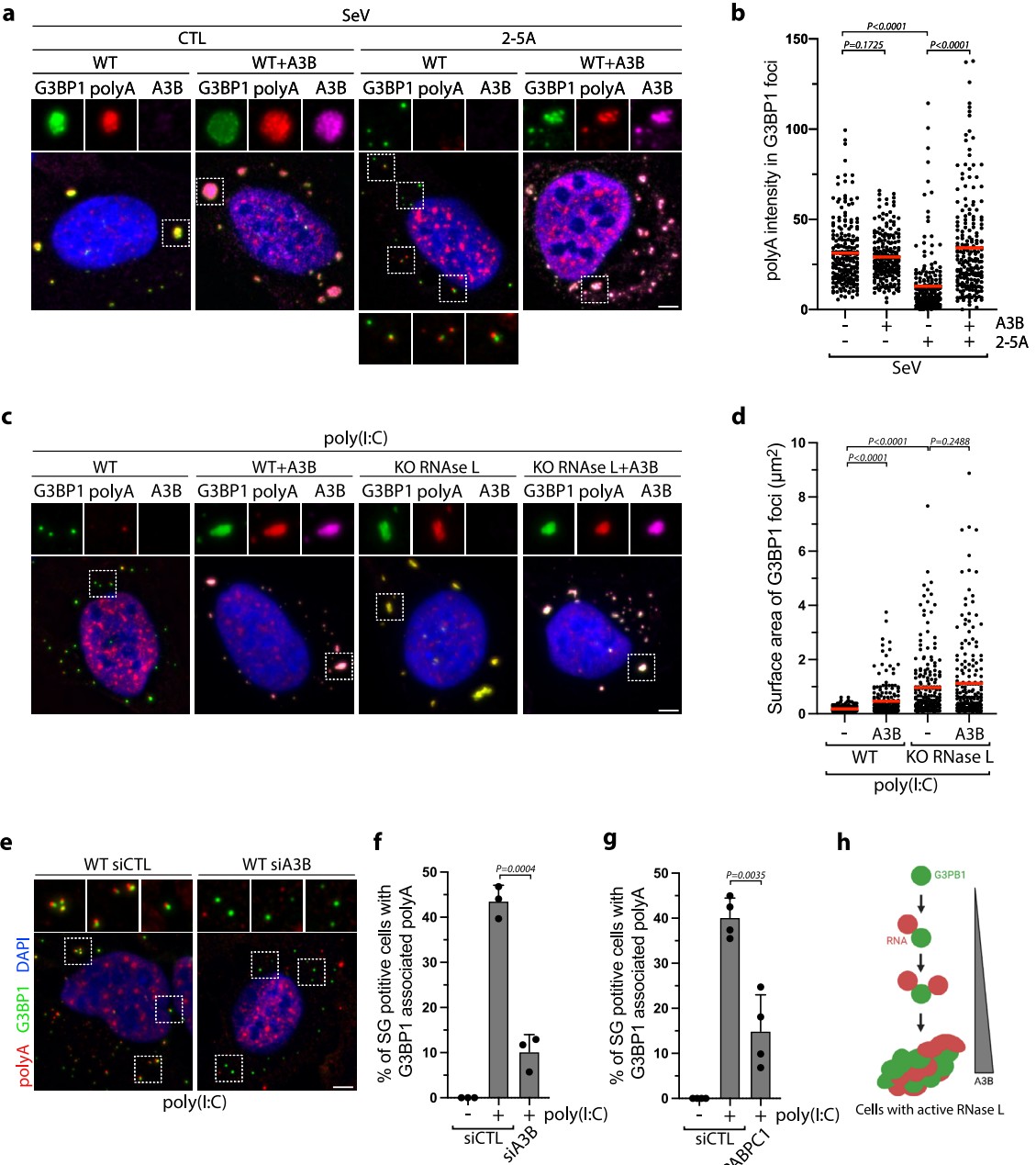

**Fig. 9 | APOBEC3B counteracts RNAse L from limiting G3BP1-RNA condensate.**
**a** U2OS-A3B-flag cells ±DOX were transfected with 2–5 A (20 μM) 8 h after cells were infected with SeV (MOI = 1). At 24hpi, G3BP1, poly(A), and A3B localization were monitored by sequential IF-FISH. Scale bar: 5 μm. **b** Quantification of poly(A) intensity (arbitrary units) in G3BP1 foci from the experiment described in (**a**). Red lines indicate the mean (Number of G3BP1 foci, n = 200). P-values were calculated with a two-tailed Welch t-test. **c** Localization of G3BP1, poly(A), A3B were monitored by sequential IF-FISH after poly(I:C) transfection (200 ng/mL, 16 h) in U2OS-A3B-flag and U2OS-A3B-flag-RNase L KO cells ±DOX. Scale bar: 5 μm. **d** The surface area of G3BP1 foci was quantified following poly(I:C) (200 ng/mL, 16 h) transfection of

the indicated cell lines. Red lines indicate the mean (Number of G3BP1 foci, n = 200). **e** U2OS cells were knocked down with indicated siRNAs followed by transfection with poly(I:C) (200 ng/mL, 4 h). Poly(A) and G3BP1 localization were monitored by IF-FISH. Scale bar: 5 μm. Quantification of percentage of cells with G3BP1-associated poly(A) in U2OS knocked down for A3B (Number of biological replicates, n = 3) (**f**) or PABPC1 (Number of biological replicates, n = 4) (**g**) for 40 h followed by poly(I:C) (200 ng/ mL, 4 h) transfection. Mean values ± SD. P-values were calculated with a two-tailed Welch t-test. **h** Increase of A3B levels promotes G3BP1-RNA condensates in the presence of active RNase L. Source data are provided as a Source Data file.

Supplementary Fig. 9B), further confirming the importance of A3B's mRNA protection function in the presence of active RNase L. Finally, we knocked down PABPC1, which is essential to recruit A3B to SGs (Fig. 2h). In the absence of PABPC1, mRNA localization to G3BP1-positive foci post poly(I:C) was strongly suppressed (Fig. 8g), validating that A3B is critical for the protection of mRNAs in RLSGs.

To further determine whether A3B protects mRNA-associated SGs, we infected cells with SeV followed by transfection of purified

2–5A, which induces RNase L activity in SeV-infected cells (Fig. 6h). 2–5 A promoted SG-associated RNAs disassembly post-SeV infection and restored the formation of G3BP1-poly(A) distinct condensates (Fig. 9a, b). Remarkably, A3B expression protected SG-associated RNAs by counteracting RNase L-mediated SG destabilization in SeV-infected cells, whereas A3B expression did not impact mRNA signals associated with SGs formed after SeV infection alone in WT cells or in the presence of 2–5 A in RNase L KO cells (Fig. 9a, b, and Supplementary

Fig. 9C). Moreover, mRNA levels remained abundant and localized to SGs in RNase L KO cells post poly(I:C) and were not affected by A3B expression (Fig. 9c, d). Together, these results demonstrated that A3B protects RNAs present in SGs against RNase L activity to maintain the structural organization of SGs.

Finally, we determined the impact of endogenous A3B on mRNAs associated with G3BP1. We focused on the punctate poly(A) signals visible in closely proximity to punctate G3BP1 foci (Fig. 9e). Interestingly, we first found that neither G3BP1 nor poly(A) punctate foci colocalized with P-bodies, which are known to interact with RLBs[57]. Instead, G3BP1, poly(A), and P-bodies were closely associated yet formed distinct condensates (Supplementary Fig. 9D). However, when we knocked down endogenous A3B, this close association was abrogated. (Fig. 9e, f). Likewise, A3B knockdown strongly decreased G3BP1 and poly(A) association in PKR KO cells, confirming that the formation of these distinct G3BP1-RNA condensates was mediated through RNase L activity (Supplementary Fig. 9E); similar results were obtained after SINV infection (Supplementary Fig. 9F). Consistent with the observation that PABPC1 is essential for the recruitment of A3B with G3BP1 foci, PABPC1 knockdown also abrogated the association between G3BP1 and poly(A) punctate foci in WT and PKR KO cells (Fig. 9g and Supplementary Fig. 9G). Therefore, these combined results support the conclusion that A3B protects RNAs associated with G3BP1 from RNase L-induced RNA cleavage. We propose that SG assembly that is known to occur through a liquid-liquid phase separation (LLPS) process[91,92,94–96] is protected by A3B to promote G3BP1-RNA condensate formation when RNase L is triggered during viral infection (Fig. 9h).

## Discussion

APOBEC3 enzymes are important components of the innate immune system to counteract diverse viral infections that cells may encounter[2,97]. APOBEC3-mediated deamination of viral genomes blocks the production of new viruses by generating lethal mutations, leading to the loss of function of newly synthesized viral proteins and viral genome degradation[2,7,97]. In contrast, mutations induced by APOBEC3 enzymes can also drive viral evolution and the production of new viral variants with altered features resistant to cell defense mechanisms[2,7,98]. The double-edged sword of APOBEC3 deamination activity suggests that APOBEC3 enzymes may have developed other mechanisms, in parallel to inducing mutations, to inhibit viral replication and prevent viral evolution. In this study, we discovered that A3B modulates the innate immune response at different stages of viral infections and defined the specific mechanism. First, we showed that A3B promotes PKR activation-induced SG formation and translation shutdown to limit viral protein synthesis. Second, we found that A3B protects RNA-associated SGs from RNase L-mediated RNA decay to promote G3BP1-RNA condensates and SG formation (refer to the models shown in Fig. 10).

The tight regulation of PKR in cells is critical for preventing self-dsRNA-induced translation shutoff while permitting its activation when a virus actually infects cells[40]. In uninfected cells, ADAR1 edits or binds endogenous self-dsRNA to block them from activating PKR[40,42]. However, ADAR1 is often hijacked by viruses to suppress PKR activation and translation inhibition to favor viral replication[45–47]. Herein, we identified a new complex formed between A3B and PABPC1 that is essential to induce PKR activity during viral infection. We propose that the role of A3B and PABPC1 is to counterbalance ADAR1's pro-viral activity and allow PKR activation to block translation and limit viral replication. However, the mechanism by which A3B and PABPC1 stimulate PKR requires further investigation. A3B and PABPC1 RNA-binding activities may prevent host or viral RNA-binding proteins (RBPs) from coating the RNAs, rendering the RNAs more accessible for PKR binding. Canonical PKR activation requires binding to dsRNA with a minimum of 30 bp length[99]. In addition to the canonical activation by

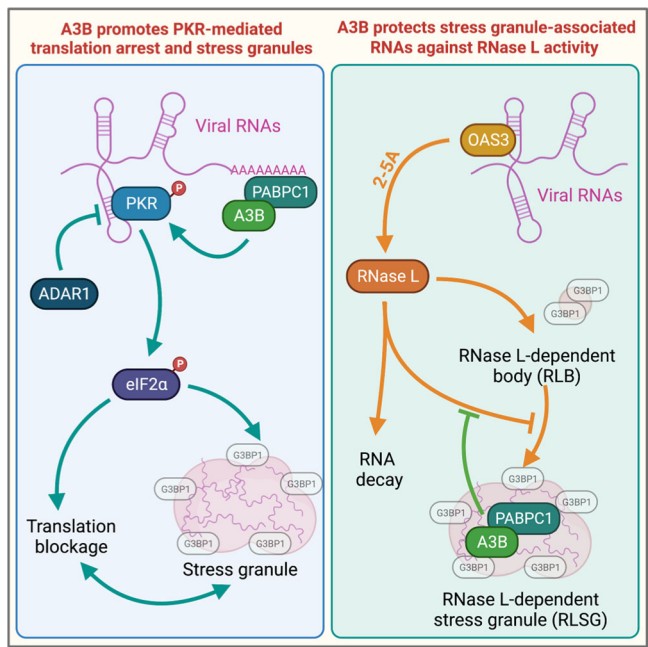

**Fig. 10 | A3B drives PKR-mediated translation arrest and protects stress granules against RNase L activity.** A3B forms a complex with PABPC1 to stimulate PKR and counterbalances the PKR-suppressing activity of ADAR1 in response to viral infection. Furthermore, A3B localizes to stress granules through the interaction with PABPC1 to protect mRNA associated with stress granules from RNAse L-induced RNA cleavage during viral infection.

dsRNA, other types of RNA substrates, such as single-stranded RNAs with limited secondary structures and with a 5′-triphosphate structure have been proposed to activate PKR[100]. Although the exact structure of these RNAs that can non-canonically activate PKR is still highly debated in the field[101,102], we speculate that A3B and PABPC1 might also make PKR binding to non-canonical RNA structures more favorable, leading to its activation. Furthermore, many virus genomic RNAs possess 5′-triphosphate structures, contrary to the host cellular transcripts that have a 7-methyl-guanosine (7mG) cap[103]. A3B and PABPC1 may also have an important function in linking 3′ poly(A) tracts to the 5′-triphosphate of viral mRNAs and thus might promote PKR activation through this mechanism. It is worth noting that numerous viruses have developed strategies to cleave or reduce PABPC1 to hijack host translation machinery. Thus, PABPC1 inhibition by viruses could also be vital to limit PKR activation and translation arrest during viral infections. More importantly, certain viruses such as the Dengue and Zika virus promote the PKR pathway to induced translation arrest while limiting SG formation to block host mRNA translation and favor their own replication through a non-canonical translation mechanism[104–107]. These observations suggest that PKR regulation by A3B may not necessarily result in a decrease in viral replication but could also be beneficial for the virus depending on the type and context. Thus, it is tempting to speculate that A3B could act as a pro-viral protein for the replication of Dengue and Zika viruses, and it will be critical in the future to precisely determine the impact of A3B on the replication of different types of viruses.

The consequences of the observed interaction between DDX3 and A3B are still unclear. DDX3 belongs to the DEAD-box helicase family and has both antiviral and pro-viral functions depending on the type of virus infecting the cells[71]. DDX3 has been reported to be an essential factor for the replication of HIV-1, West Nile virus, Dengue virus, arenaviruses, and hepatitis C virus[71,108]. Therefore, A3B association with DDX3 may prevent the hijacking of DDX3 by these viruses for replication through the regulation of DDX3 RNA helicase activity or

localization in host cells. In contrast, DDX3 shows antiviral activity against influenza A virus through its interaction with viral proteins NS1 and NP[109]. A3B may favor the formation of a high molecular complex between DDX3 and viral proteins to inhibit viral function(s). We cannot exclude the possibility that A3B could have both antiviral and pro-viral activity depending on the context and could enhance DDX3's pro-viral role during certain types of viral infections. However, further investigations are required to determine how A3B regulates DDX3 activity to promote or limit viral replication/infectivity.

Paradoxically, RNase L activation suppresses viral replication by shutting off the translation machinery but also by disturbing SG formation, which is an important mechanism of defense against viral replication that appears to be counterproductive for cells. Herein, we demonstrate that A3B recruitment to SGs is critical in preventing RNase L-mediated SG dissociation and precluding an activity that may decrease the cells' effectiveness to suppress viral replication. Similar to PKR, RNase L activation during viral infection suppresses translation but does so by degrading viral and cellular mRNAs[36–38]. However, many mRNAs coding for antiviral proteins escape RNase L-mediated mRNA decay by evolving sequences and structures to limit their cleavage by RNase L[36,54]. In addition, our results suggest that the protection of SG-associated RNAs by A3B may be another pathway for the cells to protect specific mRNA from RNase L degradation and create a reservoir of RNAs to use when the virus has been cleared. But how does A3B protect SG-associated mRNAs? The A3B N-terminal domain binds single-stranded RNAs[61,75] and may be important not only for A3B recruitment to SGs, but also for promoting a network of RNA-RNA, protein-RNA, and protein-protein interactions. SGs assemble by liquid-liquid phase separation (LLPS), which occurs when soluble protein-laden RNAs coalesce into a condensed state[91,92,94–96]. A3B interactions with both SG-associated proteins and RNA may favor protein and RNA condensation in a specific conformation that makes SG-associated mRNAs inaccessible for RNase L-mediated cleavage.

A3B is structurally similar to other members of the APOBEC3 family, raising the possibility of a conserved function among other APOBEC3 enzymes to promote PKR activity and/or protect SGs from RNase L. Indeed, previous studies have reported the localization of A3C, A3F, and A3G to SGs[110–112]. APOBEC3 proteins are composed of one (A3A, A3C, and A3H) or two (A3B, A3D, A3F, and A3G) catalytic deaminase domains (CDs)[113]. CDs contains a characteristic signature motif His-X-Glu-X$_{23-28}$-Pro-Cys-X$_{2-4}$-Cys[113] and are organized in three distinct groups, named zinc (Z)-coordinating domains, Z1, Z2, and Z3 based on their homology[113]. A3B, A3C, A3F, and A3G contain one or two Z2-related CD domains. Thus, the Z2 structure configuration may play a crucial role in the recruitment of SGs and their resulting function. Consistently, we found that A3A with high homology to the C-terminal domain of A3B (Z1 type) is not recruited to SGs (Fig. 1e). However, future studies will be necessary to define how the structural characteristics of the Z2-related CD domains may be important for APOBEC localization to SGs and impact on viral replication.

The role of the other APOBEC3 in SGs is still unknown. A3C interacts with LINE-1 protein ORF1p, and both colocalize with SGs[112]. While the interaction between A3C and ORF1p is important for LINE-1 restriction[112], it is still unclear if the recruitment of A3C to SGs is essential for its antiviral function. In contrast to A3B, A3G has been found to be indirectly associated with PABPC1 and MOV10 in SGs in an RNA-dependent manner[110,111,114]. However, in SGs, A3G interacts and colocalizes with Argonaute 1 and Argonaute 2[110], suggesting that A3G may have functions restricting viruses in SGs that are distinct from or complementary to those mediated by A3B. In light of these prior studies and our current results, we propose that multiple APOBEC3s may act in SGs to provide cells with wide-ranging defense mechanisms to protect them against a broad variety of viral infections, but those specific mechanisms still need to be defined.

## Methods

### Plasmids

APOBEC3A and APOBEC3B cDNAs were synthesized by GenScript with a beta-globin intron and a Flag tag in C-terminus. The plasmids expressing APOBEC3A-GFP/Flag, APOBEC3B-GFP/Flag, APOBEC3B-ΔCTD-GFP/Flag (amino acids 1–185), and APOBEC3B-ΔNTD-GFP/Flag (amino acids 186–382) were generated by inserting the cDNA into the pcDNA-DEST53 vector using the Gateway Cloning System (Thermo Fisher Scientific). The plasmids expressing APOBEC3B-Flag or APOBEC3B-E255Q-Flag were generated by inserting the cDNA into the pInducer20 vector using the Gateway Cloning System (Thermo Fisher Scientific). pcDNA3.1(+)-hA3Bi-3xHA was generated by overlapping PCR placing the human gamma globin intron (HBG2; accession M91037.1) between the natural boundaries of A3B exon 5 and exon 6 followed by ligation into a HindIII/XhoI digested pcDNA3.1(+) backbone upstream of in-frame carboxy-terminal 3xHA tags. All the APOBEC3B mutants were constructed by site-directed mutagenesis.

### Cell culture

U2OS, HEK-293T, Vero, and LLC-MK2 cells were cultured in DMEM supplemented with 10% FBS, 1% L-Glutamine, and 1% penicillin/streptomycin. A549 cells were maintained in DMEM/F12 GlutaMAX™-I supplemented with 10% FBS and 1% penicillin/streptomycin. Cell lines were purchased from either ATCC or Sigma-Aldrich. SKBR3 was maintained in McCoy's 5A supplemented with 10% FBS and 1% penicillin/streptomycin. U2OS- or A549-derived cell lines were generated by infecting U2OS or A549 cells with lentivirus expressing APOBEC3B under a doxycycline-inducible promoter (pInducer20) and selected with G418 (850 μg/mL). U2OS or A549 cells were treated with doxycycline (DOX, 600 ng/mL 16-24 h prior other treatments) to induce protein expression.

### Viruses

Sendai virus (SeV) Cantell strain was purchased from ATCC (#VR-907). SeV viral titer was determined by plaque assay using LLC-MK2 cells. Cells were plated and infected with SeV in 200 μL of serum-free medium for 1 h for adsorption. Then, LLC-MK2 cells were overlaid with agarose (0.45% in culture media supplemented with 5 μg/mL acetylated trypsin). After 5 days, cells were fixed with trichloroacetic acid (10%) for 30 min, stained with crystal violet (0.1% crystal violet / 25% EtOH) for 5 min, and counted to determine viral titer. Sindbis virus (SINV) Ar-339 strain was purchased from ATCC (#VR-1585). SINV viral titer was determined by plaque assay using Vero cells. Cells were plated into 6 well dished and the adsorption of the virus was performed for 1 h at 37 °C in 200 μL of virus diluted in serum-free DMEM. Then, Vero cells were overlaid with agarose (0.45% in culture media). Poliovirus type 1 (PV) (Mahoney strain) was obtained from Dr. Eckard Wimmer at Stony Brook University, New York. PV viral titer was determined by plaque assay using HeLa cells. Cells were plated into 6 well dished and adsorption of the virus was performed for 30 min at room temperature in 200 μL of virus diluted in serum-free DMEM. Then, cells were overlaid with agarose (0.45% in culture media).

### Viral infection

U2OS or A549 cells were infected with SeV, SINV, or PV in serum-free medium at indicated MOI (Multiplicity of infection) at 37 °C for 1 h (SeV and SINV) or at 22–25 °C for 30 min (PV) for adsorption. Culture medium supplemented with 10% FBS and 1% penicillin/streptomycin was added post-adsorption.

### Cell treatment

Poly(I:C)-HMW (#tlrl-pic) and poly(I:C)-FITC (#tlrl-picf) were purchased from InvivoGen and was transfected by forward transfection with Lipofectamine 2000 (Thermo Fisher Scientific) according to the manufacturer's instructions. If not indicated otherwise, cells were

treated with Poly(I:C)-HMW at 200 ng/mL for 16 h. Purified 2−5A was transfected with Lipofectamine 2000 (Thermo Fisher Scientific) according to the manufacturer's instructions. For Surface sensing of translation (SUnSET) Assay[85], U2OS or A549 cells were treated with puromycin (10 μg/mL, MP Biomedicals #ICN10055210) for 10 min before fixation for analysis by immunofluorescence with an antibody against puromycin. Sodium arsenite (Sigma, #S7400) and $H_2O_2$ (Swan, #NDC 0869-0871-43) were added directly to the cells at the indicated concentration and time.

### RNA interference

siRNA transfections were performed by reverse transfection with Lipofectamine RNAiMax (Thermo Fisher Scientific). siRNAs were purchased from Thermo Fisher Scientific (Silencer Select siRNA). Cells were transfected with siRNA for 40 h before indicated treatment (4−8 nM). The sequences of the siRNAs used in this study are listed in Supplementary Data 1.

### CRISPR-Cas9 knockout cells

PKR, RNase L, and G3BP1 CRISPR-Cas9 knockout U2OS cell lines were performed by transfection with Lipofectamine CRISPRMAX of True-Guide Synthetic CRISPR gRNA and TrueCut Cas9 Protein v2 according to the manufacturer's instructions (Thermo Fisher Scientific). CRISPR gene editing efficiency was verified using GeneArt Genomic Cleavage Detection kit (A24372; Thermo Fisher Scientific). A3B knockout cell lines were performed by transfection of the pSpCas9(BB)−2A-Puro (PX459) plasmid containing A3B gRNAs with FuGENE 6 Transfection Reagent (E2691; Promega). Sixteen hours after transfection, cells were selected with puromycin (1 μg/mL) for 2 days. For every target, three or more independent clones were generated. gRNA sequences used in this study are listed in Supplementary Data 2.

### Antibodies

The antibodies used in this study are listed in Supplementary Data 3.

### Immunofluorescence

Cells were fixed with paraformaldehyde (3% paraformaldehyde and 2% sucrose in 1× PBS) for 20 min, washed twice with 1× PBS, and cells were permeabilized with a permeabilization buffer (1×PBS and 0.2% Triton X-100) for 5 min. Subsequently, cells were washed twice with 1×PBS, and blocked in PBS-T (1× PBS and 0.05% Tween-20) containing 2% BSA and 10% milk for 1 h. Cells were then incubated with the primary antibody diluted in 1× PBS containing 2% BSA and 10% milk at room temperature for 2 h. Coverslips were washed three times with PBS-T before incubation (1 h) with the appropriate secondary antibodies conjugated to fluorophores (Alexa-488 or Cy3). After three washes with PBS-T, cells were stained with DAPI (5 μg/mL, MilliporeSigma #D9542), and the coverslips were mounted using slow-fade mounting media (Thermo Fisher Scientific, # S36936). Images were captured using a Leica DMi8 THUNDER microscope.

### RNA fluorescence in situ hybridization (RNA FISH)

Cells were fixed with paraformaldehyde (3% paraformaldehyde and 2% sucrose in 1×PBS nuclease free) for 10 min, washed twice with 1×PBS, and cells were permeabilized with a permeabilization buffer (1xPBS and 0.2% Triton X-100) for 5 min. Subsequently, cells were washed twice with nuclease free 1×PBS and then incubated with the primary antibody diluted in nuclease free 1×PBS containing 2% BSA at room temperature for 2 h. Coverslips were washed three times with PBS-T before incubation (1 h) with the appropriate secondary antibodies conjugated to fluorophores. The stained cells were washed three times with PBS-T and then fixed again with paraformaldehyde (3% paraformaldehyde and 2% sucrose in nuclease free 1×PBS) for 10 min followed by 1 h incubation in 70% ethanol at room temperature. Next, cells were washed in FISH wash buffer (10% formamide and 2X SSC

(Saline Sodium Citrate)) for 5 min. The poly(dT) FISH probe (12.5 μM stock) diluted 1:100 in FISH hybridization buffer (10% formamide, 2X SSC, and 10% (wt/vol) dextran sulfate) was added to each cover slip placed in a petri dish containing wet paper towel and then covered with parafilm to create a humidity chamber. Hybridization was performed at 37 °C for 16 h. The cells were then washed once in FISH wash buffer for 30 min. Finally, cells were stained with DAPI (5 μg/mL, MilliporeSigma #D9542) and then incubated in 2X SSC for 5 min before the coverslips were mounted using slow-fade mounting media (Thermo Fisher Scientific, #S36936). Images were captured using a Leica DMi8 THUNDER microscope. The poly(dT) FISH probe used in this study was: 5′-Cy3-oligo (dT)30 (Integrated DNA Technologies) and AHNAK(Biosearch technologies, #SS659862-01-43)

### Immunoprecipitation

U2OS or HEK-293T cells were transfected with the indicated A3B or A3A constructs. At 24 h after transfection, cells were collected and resuspended in lysis buffer (50 mM Tris−HCl, pH 7.5, 150 mM NaCl, 1 mM EDTA, and 0.5% Igepal) containing protease inhibitors (Sigma, #P8340) and phosphatase inhibitors [NaF (5 mM) and Na3VO4 (1 mM)], incubated for 5 min on ice, and lysed by sonication. Insoluble materials were removed by high-speed centrifugation at $16,000 \times g$ for 10 min at 4 °C. Then 200 μg/mL RNase A (Invitrogen, #12091-021) and 250U of Benzonase (Sigma, #E1014) were added to the supernatant and incubated at 4 °C for 20 min. Additional insoluble materials resulting from RNA and DNA digestion were removed by high-speed centrifugation at $16,000 \times g$ for 10 min at 4 °C. Each pulldown/immunoprecipitation was carried out using soluble protein extract in 1 mL of lysis buffer. For Flag- or GFP-pulldown, 50 μL of anti-Flag M2 affinity gel (MilliporeSigma, #A2220) or 20 μL of GFP-Trap agarose beads (Chromotek, #gta-20) were added and incubated for 2 h at 4 °C. Beads were washed three times with washing buffer (50 mM Tris−HCl pH 7.5, 250 mM NaCl, 0.5% NP-40) and proteins were eluted with 80 μL of Laemmli buffer. Proteins were visualized by Western blotting using the indicated antibodies. For DDX3 and PABPC1 immunoprecipitation, 4 μg of antibody against PABPC1 (Abcam, #21060) or DDX3 (Santa Cruz, #sc81247) were added and incubated for 1 h 30 min at 4 °C. Then 40 μL agarose beads conjugated with protein G (Thermo Fisher Scientific, #22851) were added and incubated for 1 h 30 min at 4 °C. Beads were washed three times with washing buffer (50 mM Tris−HCl pH 7.5, 250 mM NaCl, 0.5% NP-40) and proteins were eluted with 50 μL of Laemmli buffer.

### Bioanalyzer

Total RNA was extracted from cells using Quick-RNA MiniPrep Kit (Zymo Research, # R1055) according to the manufacturer's instructions. The amount of total RNA was measured by NanoDrop (Thermo Fisher Scientific) and then diluted to 5 ng/μL. Then RNA samples were run on Bioanalyzer 2100 (Agilent) using the Eukaryote Total RNA Pico kit (Agilent, # 5067-1513).

### Quantitative reverse transcription PCR (RT-qPCR)

Total RNA was extracted from cells using Quick-RNA MiniPrep Kit (Zymo Research) according to the manufacturer's instructions. Following extraction, total RNA was reverse transcribed using the High Capacity cDNA Reverse Transcription Kit (Thermo Fisher Scientific). RT products were analyzed by real-time qPCR using SYBR Green (PowerUp SYBR Green Master Mix, Thermo Fisher Scientific) in a QuantStudio 3 Real-Time PCR detection system (Thermo Fisher Scientific). For each sample tested, the levels of indicated mRNA were normalized to the levels of Actin mRNA. The sequences of the PCR primers used in this study are listed in Supplementary Data 4.

### Purification of OAS1

Human His-OAS1p42 purification was performed essentially as previously described[88]. Briefly, the pET9d-His-OAS1p42 plasmid was

transformed into E.coli BL21 (DE3) and IPTG was added to induce protein expression. The cells were lysed in B-PER Protein Extraction Reagent (Thermo Scientific). The cleared lysates were diluted and protein purification was done with a HiTrap Chelating HP column (GE Healthcare). The His-OAS1p42 was eluted with imidazole, and the eluted fractions were finally mixed with one volume of 80% glycerol and stored at −80 °C.

## Synthesis and purification of 2−5A

2−5A synthesis and purification were performed essentially as previously described[88]. Briefly, in 200 mL reactions, 17 µg/mL of human His-OAS1p42 was incubated with 2 mM ATP, 4 mM Mg(OAc)$_2$, 0.2 mM DTT, 40 µM EDTA, 0.2 mg/mL Poly(I:C), 0.1 mg/mL BSA, 2% glycerol and 4 mM Tris-Cl pH 7.8 for 3 h at 37 °C. In the time span 20−90 min, ATP was added every 10 min to sequentially increase the ATP concentration by 1 mM ending with a final concentration of 10 mM. The reaction sample was heat-inactivated. Fractionation and purification of 2−5As were done with successive rounds of Mono Q HR 16/10 chromatography. Mass spectrometry was used for the exact identification of the isolated 2′−5′ oligoadenylates. The trimer pppA(pA)$_2$ and/or the tetramer pppA(pA)$_3$ were used.

## Mass spectrometry

HEK-293T cells were transfected with the indicated constructs. At 24 h after transfection, cells were collected and resuspended in lysis buffer (50 mM Tris−HCl, pH 7.5, 150 mM NaCl, 1 mM EDTA, and 0.5% Igepal) containing protease inhibitors (Sigma, #P8340) and phosphatase inhibitors [NaF (5 mM) and Na3VO4 (1 mM)], incubated for 5 min on ice, and lysed by sonication. Insoluble materials were removed by high-speed centrifugation at 16,000 × g for 10 min at 4 °C. Then 200 µg/mL RNase A (Invitrogen, #12091-021) and 250U of Benzonase (Sigma, #E1014) were added to the supernatant and incubated at 4 °C for 20 min. Additional insoluble materials resulting from RNA and DNA digestion were removed by high-speed centrifugation at 16,000 × g for 10 min at 4 °C. Hundred microliters of anti-Flag M2 affinity gel (Sigma, #A2220) were added and incubated for 2 h at 4 °C. Beads were washed three times with washing buffer (50 mM Tris−HCl pH 7.5, 250 mM NaCl, 0.5% NP-40, 1 mM EDTA, and 0.5% Igepal) and proteins were eluted twice in 1 vol of beads with lysis buffer and 500 µg/mL of 3X-Flag peptide (Invitrogen) for 2 h at 4 °C. Fifty microliters of GFP-Trap agarose beads (Chromotek, #gta-20) were added and incubated for 2 h at 4 °C. Beads were washed three times with washing buffer (50 mM Tris−HCl pH 7.5, 250 mM NaCl, 0.5% NP-40) and proteins were eluted with 50 µL of Laemmli buffer. Immunoprecipitated A3B was subjected to SDS-PAGE, gel bands were cut and protein was in-gel reduced, alkylated with iodoacetamide, and digested with trypsin as described previously[115]. Protein identification was performed by the Taplin Mass Spectrometry Facility at Harvard Medical School using a microcapillary LC-MS/MS mass spectrometer.

## RNA sequencing and data analysis

U2OS cells were infected with Sendai virus following indicated treatment and total RNA was extracted using the Quick-RNA MiniPrep Kit (Zymo Cat#11−328). One microgram of total RNA for each sample was used to construct the sequencing library with the Collibri Stranded RNA Library Prep Kit Illumina™ with Collibri™ H/M/R rRNA Depletion Kit (Thermo Fisher Cat#A38110096) according to the manufacturer's instructions. Final libraries were amplified using 11 PCR cycles. Library size distributions were measured using a BioAnalyzer and quantified via qPCR. Libraries were sequenced on a Novaseq 6000 platform (UCI Genomics High-Throughput Facility (GHTF)). For RNA-seq data analysis, the fastq were mapped to the Sendai virus genome using BWA mem, PCR duplicates were removed using Picard tools, (Bioproject Accession: PRJNA897945), and nucleotide variants were obtained using GATK's Mutect2.

## Statistics and reproducibility

All western blots (Figs. 1h, 2d−e, 3f−l, 5a−d, 5g, 5j−l; Supplementary Figs. 1a, 1f, g, 2b−f, 3a−d, 4a−d, 4g, h, 4l, 5a, c, e−h, 6a, 8a, d), immunofluorescences (Figs. 1a−g, 1i−k, 2c, f−i, 3a−e, 4a, 5h, 6a−d, f, g, i, j, 7a−h, 8a−g, and 9a−e; Supplementary Figs. 1b−e, 3e−h, 4e, 4j, k, 5b, d, i, 6b, 7a, b, 7e−g, 8c, 8e−h, 9b−d) and bioanalyzer (Fig. 6e, h; Supplementary Figs. 7c, d, 8b, 9a) panels were repeated at least three times and representative images were shown in this paper.

## Reporting summary

Further information on research design is available in the Nature Portfolio Reporting Summary linked to this article.

## Data availability

The RNA sequencing data generated in this study have been deposited in the NCBI Sequence Read Archive, using the Bioproject Accession: PRJNA897945. Source data are provided with this paper.

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

## Acknowledgements

We thank Michael Carpenter for his technical assistance, Matthew Weitzman laboratory (University of Pennsylvania) for helpful discussion, and Andrew Babbin (Massachusetts Institute of Technology) for comments on the manuscript. S.O. is supported by the Faculty Mentor Program from the University of California, Irvine and P.O. is supported by a California Institute for Regenerative Medicine (CIRM) stem cell biology training grant. Salary support for A.A.A. was provided in part by NIH T32-AI83196 from the University of Minnesota's Institute for Molecular Virology Training program for J.T.B. by NIAID F32-AI147813. This work was funded by support from the National Institutes of Health NCI R37-CA252081 (R.B.), NCI R00-CA212154 (R.B.), NIAID R01-AI155962 (B.L.S.), NIDDK DP1-DK130640 (M.S.), NIAID R37-AIO64046 (R.S.H.), and NCI P01-CA234228 (R.S.H.). R.S.H. is an Investigator of the Howard Hughes Medical Institute and the Ewing Halsell President's Council Distinguished Chair. R.B. is supported by the Concern foundation and a University of California Cancer Research Coordinating Committee (CRCC) award grant. This work was also made possible, in part, through access to the Genomics High Throughput Facility Shared Resource of the Chao Family Comprehensive Cancer Center Support Grant (P30-CA062203). Cartoons in Figs. 7I, 9H, and 10 were created with BioRender.com

## Author contributions

L.M., S.O., P.O., A.B., E.B., A.S. and R.B. performed all the experiments. S.O. and M.S. analyzed RNA sequencing results. P.M.M. provided purified 2–5A molecules. A.A.A., J.T.B. and R.S.H. provided key reagents. B.L.S. and R.B. supervised the design of the experiments. R.B. conceived the study and oversaw the project. R.B. wrote the paper and all authors contributed to manuscript revisions.

## Competing interests

R.B. has served as a compensated consultant for Pfizer and Health Advances. The other authors have no competing interests to declare.
