## [Peer Review File · Nature Communications]

APOBEC3B drives PKR-mediated translation shutdown and protects stress granules in response to viral infectionEditorial Note: Parts of this Peer Review File have been redacted as indicated to maintain the confidentiality of unpublished data.

REVIEWER COMMENTS

Reviewer #1 (Remarks to the Author):

The manuscript by Manjunath et al. investigates the function of newly identified protein interactors of APOBEC3B. The data show that APOBEC3B through an interaction with PABPC1 can facilitate the formation of and stabilization of stress granules that results from stimulation of PKR activity stopping translation. A3B colocalizes to stress granules through the interaction with PABPC1. The conclusions are that A3B is a regulator of the innate immune response and can impact virus replication without viral genome editing/mutagenesis. This manuscript identifies a novel interaction with A3B and a novel function of A3B that has the potential to be of significance to multiple innate immunity fields. However, the manuscript stops short of identifying a mechanism, which this reviewer feels is important since the data suggest a major role for A3B in innate immune regulation, the data challenge the existence of the singular canonical pathway of PKR regulation, and the suggested mechanisms in the discussion are testable or are not supported by the data in the manuscript.

Comments:

1. Abstract, introduction, discussion referring to APOBEC3B causing virus evolution. There is some evidence of A3B mutations being present in HPV, polyomavirus, VSV and that this may cause evolution beneficial for the virus, but there is no strong cause and effect. Further, these are DNA viruses and this manuscript assesses RNA viruses, which A3B has not previously shown to restrict directly, so I find these types of statements misleading and should be removed from the abstract. In the introduction, there are no references supporting the claim that APOBEC3 enzyme-induced mutations at lower levels promote viral evolution, these should be included. However, a caveat of this is that most of the studies were done with HIV and A3G. A3B does not restrict HIV, so it is likely not a good example to use. Furthermore, there are several studies that have experiments set up to promote growth of drug resistant variants, use only one antiretroviral, which is not used clinically, and use only one APOBEC3 enzyme, which is not what happens in cells e.g., PMID: 31801862 or PMID: 20660203. However, different results have been found with multiple APOBEC3s or an unbiased experimental system, e.g., PMID: 31025011 or PMID: 27186986. So really the situation is more complex than what the text portrays. The text is poorly referenced, and not specific to A3B. Thus these sections need to be carefully rewritten to mention what is specific to A3B and present both sides of the drug resistance story.

2. Introduction paragraph on deamination independent restriction. This paragraph reads as though the deamination dependent effect is the primary mode of restriction when it is not. This needs to be made clear that deamination independent effects work with deamination dependent restriction. Further, many of these mechanisms have not been supported consistently in the literature. Really the best studied case is A3G and blockage of RT either as a roadblock or by binding RT directly. The inhibition of integration is a byproduct of inhibiting RT or is actually due to deamination-induced degradation (not deamination independent), so I would clarify many misleading statements in this paragraph. In addition, some statements are not supported by references and the idea that APOBEC3-mediated inhibition of HBV is strictly deamination independent is out of date, see PMID: 35438620, PMID: 28637869.

3. The rationale statement, "To determine the mechanism by which A3B participates in the innate immune defense without editing viral genomes, we monitored A3B localization in cells infected with RNA viruses that cannot be directly edited by A3B-induced DNA deaminase activity." may not be correct. An author of this manuscript has recently published a pre-print stating that A3B can edit RNA, <https://www.biorxiv.org/content/10.1101/2022.06.01.494353v1>. Since A3B is localized to the cytoplasm during RNA virus replication and has been recently shown to edit RNA, there is no evidence to support that "RNA viruses that cannot be directly edited by A3B-induced DNA deaminase activity". On the contrary, the authors should check if the virus RNA is edited by A3B. From what I observed the authors showed that the E255Q A3B can have the same localization and protein interactions as A3B wt during SeV infection, but not the actual decrease in infectivity in the presence of A3B wt and E255Q. Thus, the rationale statement is not entirely justified.

4. Line 168: "the N-terminal domain (NTD) is known to have a strong DNA and RNA binding

affinity but has undetermined functions" is not correct. PMID: 28981865 showed that the NTD is responsible for A3B oligomerization and increases specific activity.

5. Discussion line 482, there is some evidence for APOBEC3 enzymes having mechanisms to avoid resistance, such as hetero-oligomerizing and increasing activity in presence of Vif (PMID: 31025011, PMID: 27881650).

6. A major experiment or result missing from the manuscript is actually showing the mechanism by which A3B and PABPC1 increase PKR activity. There are some hypotheses mentioned in the discussion, but they are not tested. It may be important to test these since the authors are proposing what seems to potentially be a major new way to increase PKR activity in addition to the canonical stimulation by dsRNA. For example, the authors propose that A3B and PABPC1 RNA binding modify the RNA structural configuration. This is testable by EMSA or other biophysical methods. If A3B or PABPC1 can make a linear RNA form a hairpin structure, it should result in a smaller RNA spatially which would run faster on a nondenaturing gel than the linear RNA with no protein bound. Or can the authors show that A3B or PABPC1 binds dsRNA? Another hypothesis is that the 5'triphosphate is the key mechanism that A3B and PABPC1 can preferentially bind. However, of the viruses tested in this manuscript, only SeV has a 5'triphosphate. The PV as a structured 5'clover leaf RNA and SINV has a 5'guanine cap. So, this hypothesis seems incorrect just based on data in the manuscript itself.

Typographical comments:

Line 182: interaction should be interactions and lead should be led

Reviewer #2 (Remarks to the Author):

APOBEC3B (A3B) belongs to a family of DNA cytosine deaminases involved in the control of several viruses. Here, Manjunath and colleagues identify A3B as a novel regulator of protein kinase R (PKR), which acts in complex with PABPC1, and promotes translation repression and stress granule (SG) formation in response to viral infection, such as that of Sendai virus and Sindbis virus. This activity counterbalances that of the double-stranded RNA (dsRNA) adenosine deaminase ADAR1, which represses PKR. In the second part of the manuscript, the authors find 3AB to localize to SGs and propose its role in protecting SG-associated mRNAs from degradation by RNase L. The finding of 3AB-PABPC1 as a PKR positive regulatory complex is novel, highly interesting and relevant, and adds to our understanding of the complex interplay between immune and stress responses. Experiments are carefully designed, controlled, and validated using a variety of approaches, including different cell lines, different viruses or synthetic dsRNA, as well as the depletion and inducible reconstitution of A3B. The first part of the manuscript describing the impact of A3B and PABPC1 on PKR activity is robust, and the conclusions are well supported by the results. The second part proposing an impact of A3B on the protection of SGs from RNase L activity doesn't rule out at this stage an impact on the formation of SGs rather than on their dissolution as proposed by the authors. The authors need to strengthen their conclusions on this part by providing additional experimental evidence.

Specific comments:

- In general, western blot analyses are of high quality but none are quantified. This is particularly important when discussing differences in PKR phosphorylation levels, e.g. Fig. 3, Fig. 5, etc. The number of replicates is also not indicated in the legend.
- Is the impact of A3B on PKR activity the result of a direct interaction with PKR? Or indirectly through the binding to dsRNA or RNA? The pulldown performed to identify A3B interactors (Fig.2) was performed in the absence of PKR triggers. A pulldown in infected cells or cells transfected with dsRNA, treated or not with RNase A or RNase III, would clarify this point, provide better mechanistic insight, and support or revise the model presented in Fig. 10.
- ADAR1 activity was reported to destabilize dsRNA structures thereby preventing from OAS and RNaseL activation (Li et al., 2017 eLife PMID 28362255). The authors investigated the interplay between A3B-PABPC1 and ADAR1 using silencing of endogenous ADAR1 in SeV-infected cells. However, as they demonstrated in Fig. 6E, SeV infection does not induce RNase L activity.

Performing ADAR1 silencing in the context of cells transfected with dsRNA would provide the missing link between the two parts of the model shown in Fig. 10.

- Figure 6A-D. The size of G3BP1 foci in SeV-infected cells transfected with poly(I:C) increases in the presence of A3B. Although mentioned, quantifications are not provided for the 3AB-E255Q mutant.

- Fig. 6,8 and 9. The authors hypothesize and then conclude that A3B expression prevents SG dissociation caused by RNase L activity (line 376). The concept of protection from RNaseL degradation is also used. However, this statement is preliminary for this reviewer who would see another possible interpretation. The mRNAs do indeed serve as a scaffold for the formation of SGs. Their degradation before the condensation event could simply lead to the formation of smaller SGs, i.e. impact formation rather promoting dissociation. The kinetics of SeV-induced SG formation is therefore important to clarify this possibility. Fig. 6H-J, cells were infected for 8 hours before treatment with 2-5A and results analyzed at 24 hours post infection. Are SGs already formed at 8 hours post infection? An experiment in which cells are infected for 24 hours before treatment with 2-5A would be important clarify this point.

- The same question arises for the doxycycline experiments shown in Figures 8 and 9. It appears from the different legends that 3AB expression was induced by doxycycline treatment prior to infection or transfection. A comparable experiment with 3AB induction performed after SG formation will clarify the mode of action, formation vs dissolution/protection.

- Fig.9C -G, how does the silencing of A3B in the context of KO RNaseL and SeV infection +/- 2-5A impact the size and percentage of SGs?

Minor comments:

- The discussion part on further investigation with Dengue and Zika viruses (discussion line 513-516) is somewhat unclear knowing that several reports show that they block the formation of SGs.

- Although activation of PKR via the 5'-triphosphate of short-stem loops (line 502-503) has been suggested, this statement is controversial and disputed by more recent reports (Mayo et al, RNA 2016 PMID 27208315; Safran et al, RNA 2019 PMID 31239298). This part of the discussion should be revised based on these results.

- In general, the legends could benefit from better descriptions of the experiments.

Reviewer #3 (Remarks to the Author):

Using a model of Sendai Virus infection in cancer cell lines, Manjunath, Oh and colleagues show that infection results in a re-localisation of the otherwise nuclear APOBEC3B enzyme (A3B), to cytoplasmic stress granules. They also provide evidence for a novel protein complex, containing A3B, PABPC1, DDX3, and MOV10, which relocates to SGs upon SeV infection. The authors present a series of experiments using siRNA and CRISPR-Cas9-mediated gene knockouts to identify two functions for A3B in stress granules: activation of PKR and therefore repression of protein synthesis and protection of stress granules from RNase L activity. They extend their findings to other RNA viruses, including poliovirus and Sindbis virus. Both functions are shown to be independent of A3B's deaminase activity but as the authors acknowledge in their discussion, the mechanism by which A3B activates PKR remains unclear. In most cases, the extensive data presented clearly support the authors' conclusions and this study will no doubt be of considerable interest due to the new insight it affords into the innate immune response to RNA viruses and a novel role for A3B therein.

PABPC1 is required for recruitment of A3B to G3BP1 (SG component) but also for SeV-induced SG formation, so I don't think the authors can conclude that PABPC1 is required for recruitment of A3B to SGs (lines 259-260 "Thus, PABPC1 is critical for both A3B localization to SGs and the formation of SeV-induced SGs") based on this evidence alone. However, from Figure 2G (right panel), it looks as though arsenite-induced SG formation is PABPC1-independent but that the recruitment of A3B to arsenite-induced SGs is PABPC1-dependent (Figure 2F), therefore this experiment provides important support for the authors' above conclusion and I think it is worth drawing the readers' attention to it in the results text. It would also be useful to quantify the number of G3BP1+ve foci (SGs) in arsenite-treated ctrl vs siPABPC1 cells to support the interpretation of the picture shown in Figure 2G.

Related to the above point, is PABPC1 also dispensable for the induction of SGs by poly(I:C)? This is assumed in the interpretation of the experiments presented in Figure 7F, 8G and Supplementary Figure 6F, where the importance of PABPC1-mediated A3B recruitment as opposed to PABPC1-mediated SG formation, is assumed. I may have missed it (there are a lot of results shown in this manuscript!) but I think it is important to present quantification of SGs in poly(I:C)-treated ctrl vs siPABPC1 cells to convince the readers of this point.

Regarding the description of A3B's role in converting RLBs to 'unique RLSGs', could the authors comment on whether there are physiological conditions under which this may occur (i.e. is this potentially an important role for A3B in cells with active RNase-L but inactive PKR, or is this a phenomenon restricted to these experiments in which PKR is absent?) This might best be addressed in the discussion.

As with any attribution of a physiological role to A3B, the question of what happens in the significant fraction of the global population that do not express A3B protein due to the APOBEC3A_B deletion polymorphism, is worth consideration. The authors do mention potential roles for other APOBEC3 proteins in stress granules in the discussion but I would like to see this point properly investigated. One option might be to use the SKBR3 breast cancer cell line, that is homozygous for the APOBEC3A_B allele (as is SiHa (cervical cancer) and several other cancer cell lines), to assess SG formation in response to SeV infection and/or poly(I:C) and to use siRNA for the different APOBEC3 genes (and maybe other APOBECs?) to see whether one or more may be compensating for the absence of A3B in these cells. The authors have already eliminated A3A as a candidate as it does not localise to SGs, so that only leaves 5 A3 genes to test. Since (as far as I am aware) there are no epidemiological data suggesting APOBEC3A_B carriers are at increased risk from RNA virus infection, one must assume that either another gene is able to compensate for the absence of A3B in these individuals, or that this function is not of great functional importance.

Minor comments:

In Figure 2J, it would be more accurate to state that A3B partially restored the size of G3BP1 foci as opposed to stating that it "reversed the decrease of G3BP1 foci size post-2-5A transfection" (line 386).

The authors clearly show the function of A3B in SG formation is deaminase-independent, however given the recent evidence suggesting an RNA editing activity for A3B (at least when over-expressed: <https://www.biorxiv.org/content/10.1101/2022.06.01.494353v1>), it might be interesting to check whether SG-localised RNAs (e.g. AHNAK) display any evidence of A3B-mediated C>U editing under these conditions. If so, it could point to additional functions for A3B at the SG. I acknowledge that while potentially interesting, this point may be considered to be beyond the scope of the current study, so this is intended purely as a suggestion.

REVIEWER COMMENTS

Reviewer #1 (Remarks to the Author):

The manuscript by Manjunath et al. investigates the function of newly identified protein interactors of APOBEC3B. The data show that APOBEC3B through an interaction with PABPC1 can facilitate the formation of and stabilization of stress granules that results from stimulation of PKR activity stopping translation. A3B colocalizes to stress granules through the interaction with PABPC1. The conclusions are that A3B is a regulator of the innate immune response and can impact virus replication without viral genome editing/mutagenesis. This manuscript identifies a novel interaction with A3B and a novel function of A3B that has the potential to be of significance to multiple innate immunity fields. However, the manuscript stops short of identifying a mechanism, which this reviewer feels is important since the data suggest a major role for A3B in innate immune regulation, the data challenge the existence of the singular canonical pathway of PKR regulation, and the suggested mechanisms in the discussion are testable or are not supported by the data in the manuscript.

We thank the reviewer for his/her appreciation of the significance and quality of our work and for the excellent suggestions to strengthen our introduction/discussion and for pointing out missing results in our model that we have now addressed.

Comments:

1. Abstract, introduction, discussion referring to APOBEC3B causing virus evolution. There is some evidence of A3B mutations being present in HPV, polyomavirus, VSV and that this may cause evolution beneficial for the virus, but there is no strong cause and effect. Further, these are DNA viruses and this manuscript assesses RNA viruses, which A3B has not previously shown to restrict directly, so I find these types of statements misleading and should be removed from the abstract. In the introduction, there are no references supporting the claim that APOBEC3 enzyme-induced mutations at lower levels promote viral evolution, these should be included. However, a caveat of this is that most of the studies were done with HIV and A3G. A3B does not restrict HIV, so it is likely not a good example to use. Furthermore, there are several studies that have experiments set up to promote growth of drug resistant variants, use only one antiretroviral, which is not used clinically, and use only one APOBEC3 enzyme, which is not what happens in cells e.g., PMID: 31801862 or PMID: 20660203. However, different results have been found with multiple APOBEC3s or an unbiased experimental system, e.g., PMID: 31025011 or PMID: 27186986. So really the situation is more complex than what the text portrays. The text is poorly referenced, and not specific to A3B. Thus these sections need to be carefully rewritten to mention what is specific to A3B and present both sides of the drug resistance story.

We would like to apologize for oversimplifying the extended literature on APOBEC3-induced mutations and on APOBEC3 promoting viral evolution due to space limitations. As suggested by the reviewer, we have now rewritten the paragraph on APOBEC3-induced mutations and viral evolution in the introduction and added more references in the introduction and discussion to better acknowledge previously published works. Moreover, because the focus of our study is not on A3B-induced mutation and viral evolution, we only wanted to mention them as two possible outcomes. Thus, we shorten the paragraph on these related topics. In addition, we agree with the reviewer that the role of A3B in promoting viral evolution is still not well characterized, and further study will be required to determine the impact of A3B on viral evolution. Thus, we removed sentences referring to A3B causing viral evolution.

2. Introduction paragraph on deamination independent restriction. This paragraph reads as though the deamination dependent effect is the primary mode of restriction when it is not. This needs to be made clear that deamination independent effects work with deamination dependent restriction. Further, many of these mechanisms have not been supported consistently in the literature. Really the best studied case is A3G and blockage of RT either as a roadblock or by binding RT directly. The inhibition of integration is a byproduct of inhibiting RT or is actually due to deamination-induced degradation (not deamination independent), so I would clarify many misleading statements in this paragraph. In addition, some statements are not supported by references and the idea that APOBEC3-mediated inhibition of HBV is strictly deamination independent is out of date, see PMID: 35438620, PMID: 28637869.

Again, we would like to apologize for oversimplifying our description of previous literature reporting APOBEC3-mediated deamination independent function against different viruses. As mentioned by the reviewer, many proposed mechanisms have not been supported consistently in the literature further highlighting the need to better study how APOBEC3 enzymes protect cells against viruses independently of their DNA or RNA deaminase activity. Our goal was to illustrate with a few examples how little is known about the mechanisms used by APOBEC3 enzymes to limit viral replication without editing their genomes. We have now updated our statement to better reflect the current literature following the reviewer's suggestions. We also agree with the reviewer that deamination-dependent and -independent mechanisms can work together to restrict viruses. We have now illustrated this point in our introduction by citing the example of how A3G suppressing HBV can occur through both deamination-dependent and independent mechanisms. We thank the reviewer for helping us improve this part of our manuscript!

3. The rationale statement, "To determine the mechanism by which A3B participates in the innate immune defense without editing viral genomes, we monitored A3B localization in cells infected with RNA viruses that cannot be directly edited by A3B-induced DNA deaminase activity." may not be correct. An author of this manuscript has recently published a pre-print stating that A3B can edit RNA, <https://www.biorxiv.org/content/10.1101/2022.06.01.494353v1>. Since A3B is localized to the cytoplasm during RNA virus replication and has been recently shown to edit RNA, there is no evidence to support that "RNA viruses that cannot be directly edited by A3B-induced DNA deaminase activity". On the contrary, the authors should check if the virus RNA is edited by A3B. From what I observed the authors showed that the E255Q A3B can have the same localization and protein interactions as A3B wt during SeV infection, but not the actual decrease in infectivity in the presence of A3B wt and E255Q. Thus, the rationale statement is not entirely justified.

This is an important point! We agree with the reviewer that it is critical to demonstrate that A3B cannot specifically edit RNA viruses. Thus, we have now performed RNA-sequencing of cells infected with SeV and compared it to SeV-infected cells overexpressing A3B or knockdown for A3B. From these sequencing results, we were not able to significantly detect A3B-dependent RNA editing events above background level in the SeV transcriptome (**New Supplementary Figure 4M-N**) further confirming our results that A3B regulates PKR independently of its deaminase activity.

The Alonso et al. manuscript reported a few very specific RNA-editing events in certain mouse tissues overexpressing human A3B. However, to date, A3B-induced RNA editing events have not yet been reported in a human cell context. It is possible that RNA editing events described in the Alonso et al. manuscript are specific to this mouse model which artificially overexpresses human A3B or are not the direct consequences of A3B activity but arise from A3B-induced DNA damage and cell death. We previously showed that APOBEC3A is a major player in inducing RNA editing events on UpC motif in human cancer cells. Furthermore, in our study, we failed to find any correlation between RNA editing events on UpC sites and endogenous A3B expression (Jalili et al. Nature Communications, 2020). Although we cannot exclude the possibility that A3B may induce some rare RNA editing events in human cells, based on our new sequencing results, it is unlikely to have a major impact on PKR dsRNA binding activity and SG granule formation.

4. Line 168: “the N-terminal domain (NTD) is known to have a strong DNA and RNA binding affinity but has undetermined functions” is not correct. PMID: 28981865 showed that the NTD is responsible for A3B oligomerization and increases specific activity.

We agree with the reviewer and we have now modified our statement on the N-terminal domain (NTD) by: “...the N-terminal domain (NTD) is known to have a strong DNA and RNA binding affinity and facilitate A3B enzymatic activity (Xiao et al. NAR, 2017 and Adolph et al. NAR, 2017).”

5. Discussion line 482, there is some evidence for APOBEC3 enzymes having mechanisms to avoid resistance, such as hetero-oligomerizing and increasing activity in presence of Vif (PMID: 31025011, PMID: 27881650).

We have now referenced these studies in our manuscript as examples of how APOBEC enzymes can promote resistance. Thanks!

6. A major experiment or result missing from the manuscript is actually showing the mechanism by which A3B and PABPC1 increase PKR activity. There are some hypotheses mentioned in the discussion, but they are not tested. It may be important to test these since the authors are proposing what seems to potentially be a major new way to increase PKR activity in addition to the canonical stimulation by dsRNA. For example, the authors propose that A3B and PABPC1 RNA binding modify the RNA structural configuration. This is testable by EMSA or other biophysical methods. If A3B or PABPC1 can make a linear RNA form a hairpin structure, it should result in a smaller RNA spatially which would run faster on a non-denaturing gel than the linear RNA with no protein bound. Or can the authors show that A3B or PABPC1 binds dsRNA? Another hypothesis is that the 5'triphosphate is the key mechanism that A3B and PABPC1 can preferentially bind. However, of the viruses tested in this manuscript, only SeV has a 5'triphosphate. The PV as a structured 5'clover leaf RNA and SINV has a 5'guanine cap. So, this hypothesis seems incorrect just based on data in the manuscript itself.

We would like to apologize for the confusion about our proposed hypotheses mentioned in the discussion. We did not wish to propose that 5'triphosphate is the only key mechanism but one of the substrates that might be responsible for PKR activation that A3B and PABPC1 could regulate. The canonical pathway required a dsRNA of at least 30bp to allow PKR binding. However, many studies have now reported non-canonical ways to activate PKR in cells including 5'triphosphate RNAs (PMID: 18048689) and structured 5'clover leaf RNAs

(PMID: 35652592). Yet, it is still debated in the field what the exact structures of these non-canonical substrates are and how and when these non-canonical pathways induce PKR. Determining how different viruses induce PKR and which viral RNA structures are recognized by PKR during different types of viral infections will be critical in our understanding of how the innate immune system protects the cells from viruses. However, many additional studies will be required to solve all the mechanisms and our lab is actively working on uncovering them (see below).

A3B strongly binds ssRNA and to a lesser extent dsRNA (PMID: 28575276 / McCann et al. BioRxiv). However, native EMSAs with A3B lead to the accumulation of large protein/nucleic acid complexes in the wells (PMID: 28575276 / McCann et al. BioRxiv) rendering the quantification and analysis very difficult. For these reasons, it would be very challenging to determine how A3B stimulates PKR using EMSA or other biophysical methods at this stage. Although PKR regulation by A3B may occur through the modification of the RNA structural configuration, another possibility is that A3B makes the RNAs more accessible for PKR binding by preventing host or viral RNA binding proteins to coat the RNAs. Consistent with this second possibility, poly(I:C) already forms a duplex RNA when transfected in the cells and A3B is unlikely able to modify its structure. Thus, we decided to take a different approach and monitor direct PKR association with dsRNAs in cells to provide more mechanistic insight on how A3B increases PKR activity by regulating PKR's binding to dsRNAs. Very recent studies have reported the assembly of PKR clusters colocalizing with cytoplasmic dsRNAs (PMID: 35939694 and PMID: 35522180). Thus, we asked whether A3B impacts these PKR-dsRNA foci formation. We transfected cells with poly(I:C)-FITC (labeled dsRNA with a fluorophore) to monitor both phosphorylated PKR and dsRNAs and we were able to confirm previous studies that PKR forms large foci that are associated with poly(I:C) in the cytoplasm (**New Supplementary Figure 5I**). A3B knockdown decreased the number of PKR foci colocalizing with poly(I:C), while A3B overexpression promotes their formation (**New Supplementary Figure 5J and 5K**) suggesting that A3B impacts PKR association with dsRNAs. We thank the reviewer for pointing out this missing part in our manuscript and we have now revised accordingly the result and discussion sections.

[REDACTED]

[REDACTED]

Typographical comments:

Line 182: interaction should be interactions and lead should be led

Thanks! We have now corrected these typos.

Reviewer #2 (Remarks to the Author):

APOBEC3B (A3B) belongs to a family of DNA cytosine deaminases involved in the control of several viruses. Here, Manjunath and colleagues identify A3B as a novel regulator of protein kinase R (PKR), which acts in complex with PABPC1, and promotes translation repression and stress granule (SG) formation in response to viral infection, such as that of Sendai virus and Sindbis virus. This activity counterbalances that of the double-stranded RNA (dsRNA) adenosine deaminase ADAR1, which represses PKR. In the second part of the manuscript, the authors find 3AB to localize to SGs and propose its role in protecting SG-associated mRNAs from degradation by RNase L.

The finding of 3AB-PABPC1 as a PKR positive regulatory complex is novel, highly interesting and relevant, and adds to our understanding of the complex interplay between immune and stress responses. Experiments are carefully designed, controlled, and validated using a variety of approaches, including different cell lines, different viruses or synthetic dsRNA, as well as the depletion and inducible reconstitution of A3B. The first part of the manuscript describing the impact of A3B and PABPC1 on PKR activity is robust, and the conclusions are well supported by the results. The second part proposing an impact of A3B on the protection of SGs from

RNase L activity doesn't rule out at this stage an impact on the formation of SGs rather than on their dissolution as proposed by the authors. The authors need to strengthen their conclusions on this part by providing additional experimental evidence.

We thank the reviewer for his/her appreciation of the significance and quality of our work. We have now addressed all the reviewer comments to strengthen our conclusions on the impact of A3B modulating SG formation/dissolution.

Specific comments:

- In general, western blot analyses are of high quality but none are quantified. This is particularly important when discussing differences in PKR phosphorylation levels, e.g. Fig. 3, Fig. 5, etc. The number of replicates is also not indicated in the legend.

We thank the reviewer for commenting on the quality of the western blot generated by two very talented graduate students in the laboratory. To demonstrate the reproducibility of the critical data, the main results on PKR phosphorylation regulation by A3B and PABPC1 were shown repeatedly in different western blot panels across the manuscript (e.g., the phosphorylation decrease after siA3B in SeV-infected cells were shown in Figure 3G, Figure 3I, Figure 5B, and Figure 5J). In addition, we have now added fold change of PKR phosphorylation levels in the manuscript, and all the quantification of PKR phosphorylation was added in the source data documents. Finally, we added a statistic and reproducibility section mentioning that all the western blots were repeated at least three times and representative images were shown in the figures.

- Is the impact of A3B on PKR activity the result of a direct interaction with PKR? Or indirectly through the binding to dsRNA or RNA? The pulldown performed to identify A3B interactors (Fig.2) was performed in the absence of PKR triggers. A pulldown in infected cells or cells transfected with dsRNA, treated or not with RNase A or RNase III, would clarify this point, provide better mechanistic insight, and support or revise the model presented in Fig. 10.

This is an important point to clarify! We were not able to detect a significant interaction between A3B and PKR even in the absence of RNase A or after treatment with poly(I:C) to activate PKR (**Rebuttal Figure 2**). We detected very faint PKR bands by western blot in our IP experiment indicating a very weak interaction between A3B and PKR compared to the interaction observed between A3B and PABPC1. This suggests that the weak interaction between A3B and PKR is likely to be indirect or non-specific. (Note that we always observed less A3B pulldown after treatment with RNases, suggesting that A3B-RNA binding is important for the solubility of A3B when not in complex with other proteins). To provide more mechanistic insight on how A3B regulates PKR dsRNA binding, we decided to take a different approach and monitored direct PKR association with dsRNAs in cells. Very recent studies found that PKR forms clustered foci in cells colocalizing with cytoplasmic dsRNA (PMID: 35939694 and PMID: 35522180). Thus, we asked whether A3B affects the association between PKR and dsRNA directly in the cells by transfecting cells with poly(I:C)-FITC (labeled dsRNA with a fluorophore) and monitoring PKR cluster formation. We first confirmed that PKR formed large foci that are associated with poly(I:C) (**New Supplementary Figure 5I**). Remarkably, A3B knockdown decreased the number of PKR condensates, while A3B overexpression promote their formation (**New Supplementary Figure 5J and 5K**) further confirming our model that A3B impacts PKR association with dsRNAs.

Rebuttal Figure 2: Immunoprecipitation (IP) of A3B-Flag from U2OS-A3B-flag cells with or without DOX transfected with poly(I:C) (200ng/mL, 16h). IP were performed in the absence of in the presence of RNase A (200 µg/mL) and Benzonase (250U). The IP samples were analyzed by western blot using indicated antibodies.

- ADAR1 activity was reported to destabilize dsRNA structures thereby preventing from OAS and RNaseL activation (Li et al., 2017 eLife PMID 28362255). The authors investigated the interplay between A3B-PABPC1 and ADAR1 using silencing of endogenous ADAR1 in SeV-infected cells. However, as they demonstrated in Fig. 6E, SeV infection does not induce RNase L activity. Performing ADAR1 silencing in the context of cells transfected with dsRNA would provide the missing link between the two parts of the model shown in Fig. 10.

This is another important point! We found that in the absence of ADAR1, cells infected with SeV do not activate RNase L either (**Rebuttal Figure 3**). Many viruses have developed different strategies to evade RNase L activation, for example by inhibiting OASs activity or 2-5A binding to RNase L (PMID:33660566). Although it is still unknown how SeV prevents RNase L activation, this result suggests that in the absence of ADAR1, SeV can still counteract RNase L activation. This is a very different context than the results shown in Li et al., 2017 eLife (PMID 28362255), where dsRNA is induced by IFN- α treatment instead of virus infections. In this situation, there is no additional mechanism present to prevent RNase L activation from INF- α -induced dsRNAs when ADAR1 is absent.

Rebuttal Figure 3: U2OS cells were knocked down with siCTL or siADAR1 followed by infection with SeV (MOI=1, 24hpi) or transfected with poly(I:C) (200 ng/mL). Total RNAs were isolated and monitored for integrity by a bioanalyzer.

- Figure 6A-D. The size of G3BP1 foci in SeV-infected cells transfected with poly(I:C) increases in the presence of A3B. Although mentioned, quantifications are not provided for the 3AB-E255Q mutant.

Thanks for pointing out this missing result. We have now added the quantification for A3B-E255Q mutant (**New supplementary Figure 7B**). We observed the same increase of G3BP1 foci size when A3B-WT or A3B-E255Q are expressed.

- Fig. 6,8 and 9. The authors hypothesize and then conclude that A3B expression prevents SG dissociation caused by RNase L activity (line 376). The concept of protection from RNaseL degradation is also used. However, this statement is preliminary for this reviewer who would see another possible interpretation. The mRNAs do indeed serve as a scaffold for the formation of SGs. Their degradation before the condensation event could simply lead to the formation of smaller SGs, i.e. impact formation rather promoting dissociation. The kinetics of SeV-induced SG formation is therefore important to clarify this possibility. Fig. 6H-J, cells were infected for 8 hours before treatment with 2-5A and results analyzed at 24 hours post infection. Are SGs already formed at 8 hours post infection? An experiment in which cells are infected for 24 hours before treatment with 2-5A would be important clarify this point.

We agree with the reviewer that our data were insufficient to rule out at this stage an impact on the formation of SGs rather than dissolution. To improve this part of the manuscript, we have now performed the experiments suggested by the reviewer. Note that the comments on SG formation will be addressed in the next question below.

SeV infection leads to SG formation at 20-24hpi. Thus, to determine the impact of A3B on SG dissolution we now induced activation of RNase L after SGs are formed by transfecting SeV-infected cells with poly(I:C) at 24hpi. We selected poly(I:C) to activate RNase L rather than 2-5A because we reached a better transfection efficiency. A similar experimental strategy with poly(I:C) has been used to determine the impact of RNase L on SG dissolution (PMID: 31896577). We then monitored SG size at 24hpi or post-poly(I:C) transfection. We found that Poly(I:C) transfection leads to a strong decrease in SG size but A3B expression limits RNase L-mediated SG dissolution (**New supplementary Figure 7F**), further suggesting that A3B protects SG dissolution by RNase L.

- The same question arises for the doxycycline experiments shown in Figures 8 and 9. It appears from the different legends that 3AB expression was induced by doxycycline treatment prior to infection or transfection. A comparable experiment with 3AB induction performed after SG formation will clarify the mode of action, formation vs dissolution/protection.

We performed the experiment proposed by the reviewer. We first transfected cells with poly(I:C) for 4h (time required to activate RNase L and degrade RNAs [**Supplementary Figure 7C** and PMID: 31494035]) before adding DOX to induce A3B expression for 16h. We then compared SG size to the ones obtained after poly(I:C) alone or by expressing A3B prior to transfection with poly(I:C). Remarkably, when A3B is expressed after poly(I:C) transfection, we did not observe any increase in G3BP1 foci size (**New Supplementary Figure 7G**). This result suggests that once RNase L has degraded RNAs in the cells, A3B expression is no longer able to promote G3BP1-RNA condensates and form large SGs due to the lack of scaffold RNAs present in the cells. Thanks for suggesting this important experiment!

- Fig.9C -G, how does the silencing of A3B in the context of KO RNaseL and SeV infection +/- 2-5A impact the size and percentage of SGs?

In KO RNase L cells, we showed that 2-5A has no impact on SGs size after SeV infection (**Figure 6I and Supplementary Figure 7E**). However, the number of SGs and percentage of positive cells decrease after silencing A3B similarly to the results obtained with WT cells infected with SeV only (**Rebuttal Figure 4 compared to Figure 3B-D**).

Rebuttal Figure 4: RNase L KO cells were knocked down with siCTL or siA3B followed by infection with SeV (MOI=1, 24hpi) following by 2-5A transfection after 8h. The number of G3BP1 foci in each cell were monitored 24hpi. (n=1000). The red lines indicate the mean.

Minor comments:

- The discussion part on further investigation with Dengue and Zika viruses (discussion line 513-516) is somewhat unclear knowing that several reports show that they block the formation of SGs.

SG formation occurs downstream of PKR activation and many viruses such as Dengue and Zika viruses have developed mechanisms to block their formation by degrading important SG components but without affecting PKR activity levels (PMID: 23290869 and PMID: 30944179). It has been reported that both Dengue and Zika viruses utilize PKR activity to their own advantage to block host translation and promote viral translation. We have now modified the discussion accordingly. **“More importantly, certain viruses such as the Dengue and Zika virus promote PKR-induced translation arrest and limit SG formation to block host mRNA translation and favor their own replication through a non-canonical translation mechanism⁹⁵⁻⁹⁸.”**

- Although activation of PKR via the 5'-triphosphate of short-stem loops (line 502-503) has been suggested, this statement is controversial and disputed by more recent reports (Mayo et al, RNA 2016 PMID 27208315; Safran et al, RNA 2019 PMID 31239298). This part of the discussion should be revised based on these results.

We agree with the reviewer that the role of the 5'-triphosphate of short-stem loops in the activation of PKR is still debated in the field and we are currently further investigating the different mechanisms leading to PKR activation after viral infection. We have now revised accordingly our discussion by: **“In addition to the canonical activation by dsRNA, other types of RNA**

substrates, such as single-stranded RNAs with limited secondary structures and with a 5'-triphosphate have been proposed to activate PKR¹⁰⁰. Although the exact structure of these RNAs that can non-canonically activate PKR is still highly debated in the field^{101,102}, we speculate that A3B and PABPC1 might also make PKR binding to non-canonical RNA structures more favorable, leading to its activation.”

- In general, the legends could benefit from better descriptions of the experiments.

We have now expanded the figure legends to better describe the experiments.

Reviewer #3 (Remarks to the Author):

Using a model of Sendai Virus infection in cancer cell lines, Manjunath, Oh and colleagues show that infection results in a re-localisation of the otherwise nuclear APOBEC3B enzyme (A3B), to cytoplasmic stress granules. They also provide evidence for a novel protein complex, containing A3B, PABPC1, DDX3, and MOV10, which relocates to SGs upon SeV infection. The authors present a series of experiments using siRNA and CRISPR-Cas9-mediated gene knockouts to identify two functions for A3B in stress granules: activation of PKR and therefore repression of protein synthesis and protection of stress granules from RNase L activity. They extend their findings to other RNA viruses, including poliovirus and Sindbis virus. Both functions are shown to be independent of A3B's deaminase activity but as the authors acknowledge in their discussion, the mechanism by which A3B activates PKR remains unclear. In most cases, the extensive data presented clearly support the authors' conclusions and this study will no doubt be of considerable interest due to the new insight it affords into the innate immune response to RNA viruses and a novel role for A3B therein.

We thank the reviewer for his/her appreciation of the significance and quality of our work and for the excellent suggestions to strengthen it.

PABPC1 is required for recruitment of A3B to G3BP1 (SG component) but also for SeV-induced SG formation, so I don't think the authors can conclude that PABPC1 is required for recruitment of A3B to SGs (lines 259-260 “Thus, PABPC1 is critical for both A3B localization to SGs and the formation of SeV-induced SGs”) based on this evidence alone. However, from Figure 2G (right panel), it looks as though arsenite-induced SG formation is PABPC1-independent but that the recruitment of A3B to arsenite-induced SGs is PABPC1-dependent (Figure 2F), therefore this experiment provides important support for the authors' above conclusion and I think it is worth drawing the readers' attention to it in the results text. It would also be useful to quantify the number of G3BP1+ve foci (SGs) in arsenite-treated ctrl vs siPABPC1 cells to support the interpretation of the picture shown in Figure 2G.

Thanks for pointing this out and we agree with this comment. As suggested by the reviewer, we added quantification of the number of G3BP1 foci in Arsenite-treated cells in the absence of PABPC1. We are now showing that PABPC1 knockdown does not impact the formation of SGs following Arsenite treatment (**New Supplementary Figure 3G**) further confirming that PABPC1 only affects the recruitment of A3B to SGs after Arsenite treatment (**Figure 2F-G**). Thus, we have now modified our conclusion by:” **Surprisingly, in the absence of PABPC1, the formation of G3BP1 foci was abrogated in SeV infected cells but not after arsenite treatment or poly(I:C) transfection (Figure 2I and Supplementary Figures 3H-G), suggesting that PABPC1 has another important function in promoting SG formation exclusively during viral infection.**”

Related to the above point, is PABPC1 also dispensable for the induction of SGs by poly(I:C)? This is assumed in the interpretation of the experiments presented in Figure 7F, 8G and Supplementary Figure 6F, where the importance of PABPC1-mediated A3B recruitment as opposed to PABPC1-mediated SG formation, is assumed. I may have missed it (there are a lot of results shown in this manuscript!) but I think it is important to present quantification of SGs in poly(I:C)-treated ctrl vs siPABPC1 cells to convince the readers of this point.

Yes, PABPC1 is dispensable for G3BP1 foci formation post poly(I:C). As suggested by the reviewer, we have now added the quantification of G3BP1 foci number after poly(I:C) transfection to properly establish how PABPC1 affects G3BP1 foci formation. Our new result confirmed that siRNA against PABPC1 does not disturb G3BP1 foci formation after poly(I:C) treatment (**New Supplementary Figure 3H**). Importantly, these results now fully support the experiments presented in Figures 6F, 7F, 8G, and Supplementary Figure 6F (now Supp 7C). Thanks for pointing out these missing results!

Regarding the description of A3B's role in converting RLBs to 'unique RLSGs', could the authors comment on whether there are physiological conditions under which this may occur (i.e. is this potentially an important role for A3B in cells with active RNase-L but inactive PKR, or is this a phenomenon restricted to these experiments in which PKR is absent?) This might best be addressed in the discussion.

A3B's role in converting RLBs to RLSGs may be critical in protecting certain mRNAs and create a reservoir of RNA to use when the virus has been cleared. As poly(I:C) activates both the RNase L and PKR pathways, we performed experiments in a PKR KO background to specifically dissect the impact of A3B in converting RLBs to RLSGs. We have found that PKR does not impact the conversion of RLBs to RLSGs (**Figure 7B**). Many viruses have developed different strategies to limit or suppress PKR activation. For example, viruses such as vaccinia virus or influenza virus express proteins to compete with PKR for dsRNA binding and prevent PKR activation. Toscana virus triggers the proteasomal degradation of PKR and Hantavirus inhibits PKR dimerization (PMID: 34759908). Thus, our results obtained with PKR KO cells mimic many viral infections leading to PKR inhibition. However, it is important to note that regardless of whether the PKR pathway is activated or not, A3B's role remains crucial in converting RLB to RLSG to protect SG structures when RNase L is activated in cells.

As with any attribution of a physiological role to A3B, the question of what happens in the significant fraction of the global population that do not express A3B protein due to the APOBEC3A_B deletion polymorphism, is worth consideration. The authors do mention potential roles for other APOBEC3 proteins in stress granules in the discussion but I would like to see this point properly investigated. One option might be to use the SKBR3 breast cancer cell line, that is homozygous for the APOBEC3A_B allele (as is SiHa (cervical cancer) and several other cancer cell lines), to assess SG formation in response to SeV infection and/or poly(I:C) and to use siRNA for the different APOBEC3 genes (and maybe other APOBECs?) to see whether one or more may be compensating for the absence of A3B in these cells. The authors have already eliminated A3A as a candidate as it does not localise to SGs, so that only leaves 5 A3 genes to test. Since (as far as I am aware) there are no epidemiological data suggesting APOBEC3A_B carriers are at increased risk from RNA virus infection, one must assume that either another gene is able to compensate for the absence of A3B in these

individuals, or that this function is not of great functional importance.

This is another important point! As mentioned by the reviewer, to our knowledge, there is no epidemiological data showing either APOBEC3A_B carriers have increased, decreased, or had no differences in risk from RNA virus infections. Thus, because such studies have not yet been performed, it remains difficult to speculate what are the consequences of an RNA viral infection for the population that does not have A3B and whether other APOBEC members can compensate for the absence of A3B. Moreover, the outcome of the infection will also be dependent of the adaptive immunity. However, following the reviewer's suggestions, we monitored PKR activation and SG formation in the SKBR3 cell line that is homozygous for the APOBEC3A_B allele and compared the results to U2OS cells. Remarkably, we found a strong defect of PKR phosphorylation in SKBR3 cells infected with SeV, SINV, and PV, even in cells infected at different MOI (**New Supplementary Figures 4D and 5H**). Consistently, the level of SGs was also strongly suppressed in SKBR3 cells after SeV infection (**New Supplementary Figure 4E**).

We then analyzed the expression level of other APOBEC3 members in SKBR3 and U2OS cell lines. In addition to A3B, U2OS cells also expressed A3C and A3F at lower levels, whereas SKBR3 cells do not significantly express any APOBEC3 members (**New Supplementary Figure 4F**). Knockdown of A3C or A3F did not impact PKR phosphorylation level and SG formation suggesting neither A3C nor A3F impacts PKR activity after viral infection (**New Supplementary Figure 4G-K**). However, we still cannot completely rule out that other APOBEC3 members that are not expressed in these cell lines can compensate A3B functions in other cellular contexts, and further studies will be necessary to determine the activity of other APOBEC3 members. We think that such work would be beyond the scope of this current study focusing on A3B function.

Minor comments:

In Figure 2"6"J, it would be more accurate to state that A3B partially restored the size of G3BP1 foci as opposed to stating that it "reversed the decrease of G3BP1 foci size post-2-5A transfection" (line 386).

We have now modified our statement related to Figure 6J by: "**A3B expression partially restored the size of G3BP1 foci post-2-5A transfection**".

The authors clearly show the function of A3B in SG formation is deaminase-independent, however given the recent evidence suggesting an RNA editing activity for A3B (at least when over-expressed: <https://www.biorxiv.org/content/10.1101/2022.06.01.494353v1>), it might be interesting to check whether SG-localised RNAs (e.g. AHNAK) display any evidence of A3B-mediated C>U editing under these conditions. If so, it could point to additional functions for A3B at the SG. I acknowledge that while potentially interesting, this point may be considered to be beyond the scope of the current study, so this is intended purely as a suggestion.

We have now performed RNA-seq of cells infected with SeV and compared it to SeV-infected cells overexpressing A3B or knockdown for A3B. We were not able to detect any significant RNA editing events difference above background level in *AHNAK* mRNA as well as in the SeV transcriptome (**Rebuttal Figure 5 and New Supplementary Figure 4M-N**). A3B-induced RNA editing events have never been reported in a human cell context yet. Although we cannot

exclude that A3B induced some rare RNA editing events in human cells, it is unlikely to have a major impact on PKR dsRNA binding and SG granule formation. However, further studies will be necessary to determine whether A3B RNA editing has any function in human but we agree with the reviewer such work would be beyond the scope of this current study.

Rebuttal Figure 5: Right. RNA isolated from U2OS cells transfected with siCTL or siA3B followed by SeV infection (MOI=1, 24hpi) was sequenced and aligned to the human genome sequence to monitor RNA editing events in *ANHAK*. The number of mutations for each nucleotide variant across *ANHAK* mRNA was called by GATK's Mutect2 variant caller directly from aligned BAM files from RNA-Seq, where a minimal threshold was used to retain all possible sequence variants. Only point mutations present in three or more sequencing reads at a particular position and not present in all sequencing reads were selected as RNA editing events. Specific nucleotide mutations are shown on the x-axis and the ratio of A3B siRNA (left) or overexpression (right) are plotted relative to each control sample (y-axis). We note that across typical variant calling pipelines, no mutational frequency observed in these data would achieve a Bonferroni-adjusted significance threshold. Positive bar graph indicates more mutations for the specific nucleotide variant in siCTL sample while negative bar graph indicates more mutations in siA3B sample. **Left.** The number of mutations in SeV genome was determined in U2OS cells \pm DOX infected with SeV (MOI=1, 24hpi). Positive bar graph indicates more mutations for the specific nucleotide variant in control sample while negative bar graph indicates more mutations in A3B-overexpressed sample.

REVIEWERS' COMMENTS

Reviewer #1 (Remarks to the Author):

The authors have addressed all prior concerns.

Reviewer #2 (Remarks to the Author):

In the revised manuscript, Manjunath and colleagues have addressed in detail all of the questions and suggestions from each of the reviewers. With this additional work, my remaining concerns have been addressed. The overall quality of the manuscript has been improved. This work will be of considerable importance to the field.

Reviewer #3 (Remarks to the Author):

The authors have addressed my previous comments and I have no further concerns. The manuscript identifies a new role for APOBEC3B in innate immunity and the revised manuscript (revisions to all three sets of reviewer comments have been conducted quite comprehensively in my opinion), certainly gives a significantly clearer picture of mechanism by which this role is executed.

REVIEWERS' COMMENTS

Reviewer #1 (Remarks to the Author):

The authors have addressed all prior concerns.

We thank the reviewer for his/her support.

Reviewer #2 (Remarks to the Author):

In the revised manuscript, Manjunath and colleagues have addressed in detail all of the questions and suggestions from each of the reviewers. With this additional work, my remaining concerns have been addressed. The overall quality of the manuscript has been improved. This work will be of considerable importance to the field.

We thank the reviewer for his/her appreciation of the significance and quality of our work.

Reviewer #3 (Remarks to the Author):

The authors have addressed my previous comments and I have no further concerns. The manuscript identifies a new role for APOBEC3B in innate immunity and the revised manuscript (revisions to all three sets of reviewer comments have been conducted quite comprehensively in my opinion), certainly gives a significantly clearer picture of mechanism by which this role is executed.

We thank the reviewer for his/her appreciation of our work.